# Widespread increase in dynamic imbalance in the Getz region of Antarctica from 1994 to 2018

Heather L. Selley [1✉], Anna E. Hogg[2], Stephen Cornford[3], Pierre Dutrieux [4,10], Andrew Shepherd[1], Jan Wuite [5], Dana Floricioiu [6], Anders Kusk [7], Thomas Nagler [5], Lin Gilbert [8], Thomas Slater [1] & Tae-Wan Kim [9]

The Getz region of West Antarctica is losing ice at an increasing rate; however, the forcing mechanisms remain unclear. Here we use satellite observations and an ice sheet model to measure the change in ice speed and mass balance of the drainage basin over the last 25-years. Our results show a mean increase in speed of 23.8 % between 1994 and 2018, with three glaciers accelerating by over 44 %. Speedup across the Getz basin is linear, with speedup and thinning directly correlated confirming the presence of dynamic imbalance. Since 1994, 315 Gt of ice has been lost contributing 0.9 ± 0.6 mm global mean sea level, with increased loss since 2010 caused by a snowfall reduction. Overall, dynamic imbalance accounts for two thirds of the mass loss from this region of West Antarctica over the past 25-years, with a longer-term response to ocean forcing the likely driving mechanism.

[1] Centre for Polar Observation and Modelling, School of Earth and Environment, University of Leeds, Leeds, UK. [2] School of Earth and Environment, University of Leeds, Leeds, UK. [3] Department of Geography, Swansea University, Swansea, UK. [4] Lamont-Doherty Earth Observatory, Colombia University, New York, USA. [5] ENVEO IT GmbH, Innsbruck, Austria. [6] German Aerospace Centre (DLR), Remote Sensing Technology Institute, Wessling, Germany. [7] DTU Space, National Space Institute, Technical University of Denmark, Lyngby, Denmark. [8] Mullard Space Science Laboratory, Department of Space & Climate Physics, University College London, London, UK. [9] Korea Polar Research Institute, Incheon, South Korea. [10] Present address: British Antarctic Survey, Natural Environment Research Council (NERC), Cambridge, UK. ✉email: eehls@leeds.ac.uk

Over the last 25 years, the Antarctic Ice Sheet has contributed 7.6 ± 3.9 mm to global sea level[1]. Observations have shown that ice loss from Antarctica is dominated by the low-lying, marine-based sectors of West Antarctica[1–4], where glaciers in the Amundsen Sea Sector have thinned, accelerated, and grounding-lines have retreated since the 1940s[5–8]. Elsewhere, ice shelves on the Antarctic Peninsula have retreated[9] and collapsed[10,11], triggering a dynamic response as the flow of grounded ice is unbuttressed[12]. Dynamic imbalance in West Antarctica is driven by incursions of warm modified Circum-polar Deep Water (mCDW) melting the floating ice[3,13], with the interannual and long-term variability of ocean temperatures linked to atmospheric forcing associated with the El Nino-Southern Oscillation (ENSO)[14,15] and anthropogenic forcing[16], respectively. The ice sheet contribution to the global sea level budget remains the greatest uncertainty in future projections of sea level rise[17], driven in part by positive feedbacks such as the Marine Ice Sheet Instability (MISI)[18,19], and with the most extreme scenarios (>1 m by 2100) only possible through the onset of Marine Ice Cliff Instability (MICI)[20,21]. Satellite data have shown that in Antarctica the dynamic ice loss (6.3 ± 1.9 mm sea level equivalent (sle)) is 86% greater than the modest reduction in surface mass (0.9 ± 1.1 mm sle) since the 1990s[22]. However, both long-term and emerging new dynamic signals must be accurately measured in order to better understand how ice sheets will change in the future. Since 2008, 88% of the observed ice speedup has occurred on glaciers located in the Amundsen Sea, Getz and Marguerite Bay sectors[23]. However, the timing and pace of dynamic imbalance is poorly characterised in regions that are observed less frequently, and uncertainty remains about the physical mechanisms driving this change.

The Getz drainage basin lies at the coastal margin of Marie Byrd Land, and covers 10.2% (177,625 km²) of the West Antarctic Ice Sheet[8]. Ice flows from the ice sheet into the Getz Ice Shelf through 14 distinct glaciers that extend ~145 km inland, and flow at average speeds of over 500 m/year at the grounding line (Fig. 1a). The region is characterised by a 650-km-long 25- to 110-km-wide ice shelf, the eighth largest in Antarctica, which provides buttressing support to the grounded ice. While grounding line retreat has been observed since 2003 (ref. [24]), the ice shelf is anchored at its seaward margin by eight large islands and over 23 pinning points which stabilise the ice shelf calving front, resulting in a relatively small area change over the past three decades[25,26]. Despite the absence of significant area change, strong ice shelf thinning (−16.1 m/decade) has been observed since the 1990s[27], producing one of the largest sources of fresh water input to the Southern Ocean[28], more than double that of the neighbouring Amundsen Sea ice shelves[29]. The complex network of topographic rises at the ice front channel ocean currents under the sub-ice shelf cavity, driving a highly localised spatial pattern of ice thinning, with the strongest rates observed at the grounding line[30]. Spatially variable ocean forcing is expected along the Getz coastline due to its zonal extent, about half the length of the West Antarctic Ice Sheet margin, and its location between the colder Ross and warmer Amundsen Seas[25].

Over the past 30 years, the Getz drainage basin has lost ~410 Gt of ice mass[8], at a rate that has increased by over 40% since 2010 (ref. [31]). The Flood, McCuddin and Executive Committee mountain ranges provide regions of high elevation bed topography at the ice divide, producing a broadly prograde bed slope across the basin[32]. This geometry prevents inland propagation of strong ice sheet thinning[8] (Fig. 1c) and makes the region less susceptible to onset of MISI in comparison to the retrograde sloping marine-based glaciers in the neighbouring Amundsen Sea Embayment[18,19]. The pattern of snowfall is heterogeneous across the basin, with the highest rates deposited on steeply sloping topography aligned orthogonal (i.e. West to East) to the prevalent direction of atmospheric moisture flux[33]. While summer surface melt is limited to the lower elevation ice shelf[33], interannual variability in Surface Mass Balance (SMB) is driven by the Amundsen Sea Low, which accounts 40% of the surface mass and 21% of the surface melt variability[34]. Partitioning the influence of both surface mass and ice dynamic signals in Antarctica is key to understanding the atmospheric and oceanic forcing mechanisms driving recent change[35]. Studies suggest that thinning of glaciers flowing into the Getz Ice Shelf are greater than the estimated increase in ice discharge alone, indicating that surface processes are at least in part responsible for the observed thinning[8,36]. However, a multi-decadal, continuous ice velocity record is required to perform a detailed assessment of the change in ice flux. In this study, we use satellite data to measure the change in ice speed of glaciers in the Getz drainage basin from 1994 to 2018, to assess the localised pattern of dynamic imbalance in this large and complex sector of West Antarctica. Our results show that between 1994 and 2018, widespread speedup has occurred on the majority of glaciers in the Getz drainage basin of West Antarctica. A mean speed increase of 23.8% was observed, with three glaciers accelerating by over 44%. Since 1994, 315 Gt of ice has been lost, contributing 0.9 ± 0.6 mm to global sea levels. Increased ice loss since 2010 was driven by a snowfall reduction, with dynamic imbalance likely driven by a longer-term response to ocean forcing.

## Results

**Ice velocity.** Our velocity measurements show that the Getz coastline is characterised by 14 major flow units, with two glaciers reaching mean speeds of over 1 km/year in 2018, and regions of fast flow generally limited to within ~40 km of the grounding line (Fig. 1a). Nine of the 14 major glaciers are unnamed, with flow units 10 to 14 at the far West of the sector (Fig. 1a) corresponding to DeVicq, Berry, Venzke, Hull and Land Glaciers, respectively. We extracted velocity measurements along 14 flow-line transects located on the fast-flowing trunk of glaciers in the Getz study region (Figs. 1a and 2). Measurements were extracted from a 5-km diameter region located at the grounding line of each flow unit. The fastest flowing ice streams in the Getz drainage basin are located in the West of the sector, where Hull Glacier (flow unit 13) and Land Glacier (flow unit 14) flow at speeds of up to 1.4 ± 0.2 km/year (Table 1 and Supplementary Fig. 1). Slower moving ice is transported over the Executive Committee, McCuddin and Flood mountain ranges inland of Marie Byrd Land, into the Amundsen Sea via the Getz Ice Shelf through which the majority of glaciers flow. The slowest moving glaciers are found in the far East of the Getz drainage basin, where flow unit 1 located to the West of Martin Island, and flow unit 3 located to the West of Wright Island, flow at a speeds of 174 ± 12 m/year and 183 ± 14 m/year, respectively (Table 1 and Supplementary Fig. 1).

The spatial coverage of the ice velocity observations is influenced by both the density of satellite data acquisitions and the performance of the image processing techniques used to measure ice speed. In the Getz drainage basin, the absence of visible features on the ice sheet surface and the susceptibility of the snow surface to change in response to local weather events makes the ice sheet interior a particularly challenging region to measure. Our results show that in general, ice speed measurements are limited to within ~200 km of the calving front due to the combined effect of weather and the geographical coverage of the satellite images acquired across the Getz study region, with consistently poorer coverage inland (Supplementary Fig. 2). We observe the lowest spatial coverage in 1998, where velocity measurements are retrieved over 2% (3432 km²) of the Getz

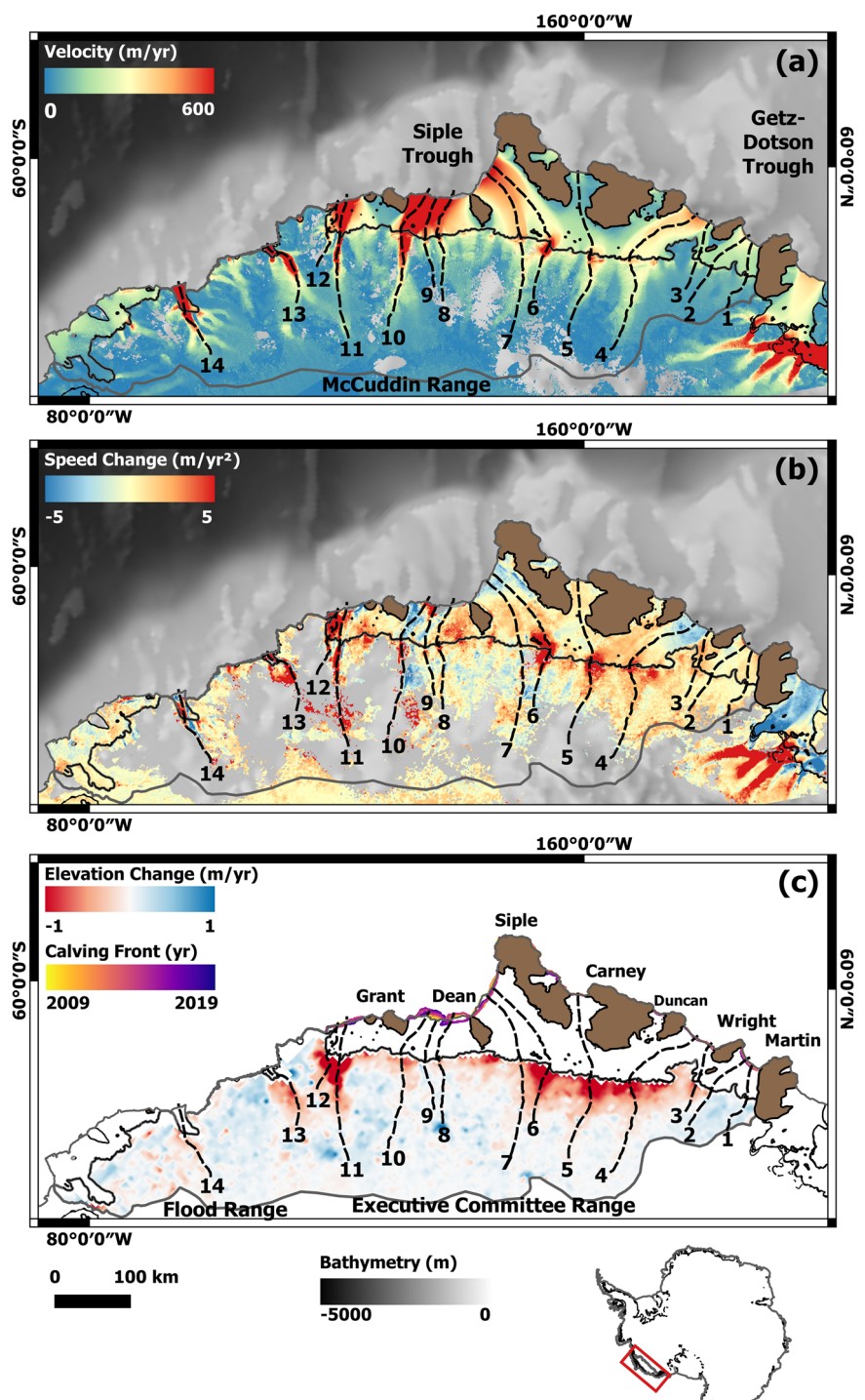

**Fig. 1 Ice speed, speed change and thinning over the Getz basin. a** Ice speed in the Getz drainage basin of Marie Byrd Land, measured in 2018 using Interferometric Wide (IW) mode synthetic aperture radar (SAR) data acquired by the Sentinel-1a/b satellites. The grounding line location (solid black line[67]), inland limit of the drainage basin (solid grey line) and the location of 14 flow line profiles (dashed black lines) are also shown. Profiles 1 to 9 are located on unnamed glaciers; however, 10 to 14 correspond to DeVicq, Berry, Venzke, Hull and Land Glaciers, respectively. Measurements are superimposed on BEDMAP2 bedrock topography[32]. **b** A map of the observed rate of change in ice speed between 1994 and 2018. **c** Ice sheet elevation change with Firn Densification Model (FDM) correction applied from 1992 to 2017, measured using satellite radar altimetry data[8]. The names of islands and ice rises (brown area) bordering the Getz Ice Shelf are also indicated. The calving front location is shown from 2009 in yellow to 2019 in purple, see Supplementary Fig. 7 for a zoom of the calving front change.

drainage basin (Supplementary Table 1). The highest spatial coverage was observed during the 2014 and 2016 epochs, where ice speed is measured over 154,209 km$^2$ (90%) and 155,518 km$^2$ (90%) of the Getz basin, respectively (Supplementary Table 1).

On average, all glaciers in the Getz drainage basin were surveyed on at least ten occasions since 1994. Though the historical observations from the 1990s have poor spatial coverage, they provide an invaluable reference measurement for glaciers in the

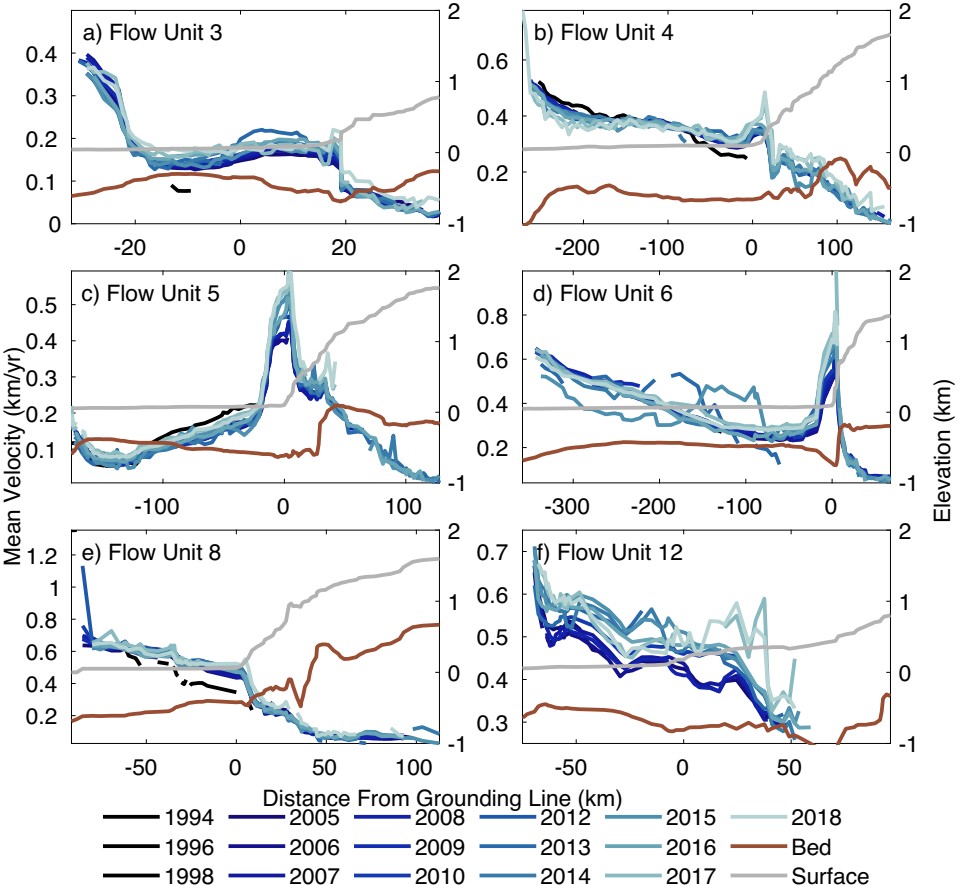

**Fig. 2 Ice speed profiles.** Profiles of ice speed observed between 1994 (dark blue) and 2018 (light blue) extracted along flow-line transects located on the central trunk of outlet glaciers in the Getz study region. The ice surface elevation (grey line) and bed elevation (brown line) from BEDMAP2 (ref. [32]) extracted along the same profile is also shown. The x-axis is shown as distance from the grounding line[67], with positive values indicating the inland section of the profile on the ice sheet, and negative values indicating seaward locations. Flow units 3 (**a**), 4 (**b**), 5 (**c**), 6 (**d**), 8 (**e**) and 12 (**f**) are presented here, with all 14 flow line profiles shown in Supplementary Fig. 8.

**Table 1 The observed mean ice speed from 2018, the observed and modelled rate of change in ice speed, the observed total and percentage change in ice speed from 1994 to 2018.**

| Flow unit | 2018 Mean speed (m/yr) | Observed rate of speed change (m/yr²) | Modelled rate of speed change (m/yr²) | Observed total speed up over 25 years (m/yr) | Observed percentage speed change over 25 years (%) | Rate of elevation change 1992–2017 (m/yr) | Grounding line migration 1996–2017 (km) | Ice thickness (m) | Bed elevation (m) |
|---|---|---|---|---|---|---|---|---|---|
| 1 | 173.7 ± 11.5 | 0.7 | 0.8 | 18.6 | 10.7 | 0.0 | −1.3 | 649 | −540 |
| 2 | 192.2 ± 17.2 | 0.2 | 0.1 | 5.5 | 2.9 | −0.2 | −1.1 | 674 | −594 |
| 3 | 182.6 ± 14.4 | 1.9 | 1.9 | 46.5 | 25.5 | −0.7 | −2.7 | 639 | −541 |
| 4 | 390.7 ± 33.3 | 3.7 | 3.4 | 92.4 | 23.6 | −1.0 | – | 775 | −651 |
| 5 | 532.4 ± 36.1 | 10.5 | 9.1 | 263.4 | 49.5 | −1.6 | – | 728 | −613 |
| 6 | 668.8 ± 90.8 | 15.6 | 15.2 | 391.2 | 58.5 | −1.8 | – | 851 | −727 |
| 7 | 424.8 ± 42.2 | 1.0 | 2.1 | 24.5 | 5.8 | −0.5 | −2.5 | 638 | −552 |
| 8 | 518.8 ± 59.1 | 2.0 | 5.4 | 50.4 | 9.7 | −0.5 | – | 501 | −418 |
| 9 | 511.2 ± 64.6 | 2.8 | 3.8 | 69.4 | 13.6 | −0.1 | – | 552 | −473 |
| 10 | 879.2 ± 149.1 | 9.1 | 11.4 | 228.2 | 26.0 | −0.6 | – | 659 | −535 |
| *11* | *744.3 ± 125.0* | *5.7* | *5.1* | *141.5* | *19.0* | *−2.2* | – | *913* | *−659* |
| 12 | 467.8 ± 65.5 | 8.2 | 6.2 | 205.5 | 43.9 | −1.3 | – | 995 | −806 |
| *13* | *1397.6 ± 163.7* | *11.4* | *6.0* | *285.8* | *20.4* | *−0.4* | – | *674* | *−524* |
| 14 | 1281.1 ± 122.6 | 8.0 | 7.3 | 198.9 | 15.5 | −0.6 | – | 701 | −560 |

All measurements were extracted from a 5-km-diameter region at the grounding line, on all 14 flow lines (Fig. 1a) in the Getz study region. Measurements of grounding line migration are provided where observations exist, along with the mean FDM corrected ice thickness and bed elevation[32] also extracted from the 5 km region at the grounding line. Ice sheet elevation change[8] was extracted from a 20-km-diameter region due to the coarser spatial resolution of this dataset. Italics denote flow units (11 and 13) with less than 70% coverage in the 5 km region on the observed speed change map. Grounding line migration is calculated as the distance along the central flow line the grounding line has moved, negative indicating a movement inland.

central to Northern part of the Getz sector, where no other ice speed estimate is available. In all other years except for 2009 and 2012, the majority (>55%) of the ice sheet margin is observed (Supplementary Table 1).

**Change in ice speed.** Our results show that the majority of glaciers in the study area have accelerated since the 1990s, with seven glaciers speeding up by over 20% (flow units 3 to 6, 10, 12 and 13) (Table 1) based on a linear rate (Supplementary Figs. 1d, 3 and 4).

In regions without observational data from the 1990s, this trend may not be tightly constrained by the data; however, we find good agreement between the modelled and observed rates (Supplementary Fig. 1). In order to asses dynamic imbalance across the Getz drainage basin over the 25-year study period, we generated a map of change in ice speed by fitting a linear trend to every pixel in the study region, for all 16 observed ice speed maps from 1994 to 2018 (Fig. 1b). We required all trends to include a minimum of 5-years of velocity observations, not necessarily consecutively, and we filtered the output using a 95% confidence interval on the fit. The largest percentage increase in ice speed was observed on flow units 5 and 6 in the centre of the Getz basin behind Siple and Carney Islands, where ice flow has increased by 50% and 59%, respectively, since 1994 (Fig. 1b and Table 1). This corresponds to two of the fastest speed change rates which are also observed on flow units 5 and 6, with speeds increasing by 10.5 m/year$^2$ and 15.6 m/year$^2$, respectively, and DeVicq Glacier (flow unit 10) in the far West, where speeds have increased by 9.1 m/year$^2$ (Table 1 and Supplementary Fig. 1d). The smallest change in speed was observed on flow units 1 and 2 in the far East of the Getz basin (Fig. 1b) where ice speeds have increased at rates of under 0.7 m/year$^2$ since 1994 (Table 1 and Supplementary Fig. 1d). While we observed the second largest speed up on flow unit 13, the spatial coverage of the speed change observations within the 5 km region at the grounding line is 40%, substantially lower than the >85% coverage on all 12 other ice streams. Consequently, we excluded this glacier, and Berry Glacier (flow unit 11) (70%), from further detailed analysis to avoid overinterpreting results in regions of poorer coverage.

We find that the glaciers that have sped up the most correspond to the regions of strongest ice sheet thinning, greater than −1.0 m/year (Fig. 1c and Table 1), which in some cases extends up to 75 km inland (flow units 5, 6, 11 and 12). We directly compared the rate of relative ice speedup (%) with the amount of ice sheet thickness change as a proportion of the total ice thickness, and found a linear relationship between the

two parameters ($R^2 = 0.74$) as required by the theory of mass conservation[37] (Fig. 3a). This result indicates that an ~50% increase in ice speed will cause an ~5% decrease in glacier thickness due to ice dynamic processes, and suggests that there is no significant bias in the velocity or surface elevation change observations.

**Optimised ice flow model**. We employed an optimised ice flow model to complement the satellite measurements by filling in gaps in the 16 annual velocity maps, and to investigate the processes driving change in ice speed. We find that the observed and modelled pattern of speed change agrees well (Supplementary Fig. 5a), with a mean difference of 0.18 m/year$^2$. The results show the number and distribution of flow units has remained uniform over the 25-year study period, the largest area of speedup is observed in the central section of the Getz drainage basin (Fig. 1b). The two spatially extensive but distinct regions of speedup, ~300 m/year since 1994, are observed at the grounding lines of flow units 5 and 6. However, in both cases, this increase is concentrated around the fastest flowing grounded ice, and does not extend to the ice shelf calving front. Strong localised speedup, of over 90 m/year since 1994, is observed on six other glaciers in the Getz basin, including flow unit 4, along with DeVicq, Berry, Venzke, Hull and Land Glaciers (flow units 10, 11, 12, 13 and 14, respectively) (Table 1). The observed speedup is again strongest at the grounding line, and in all cases apart from Venzke Glacier (flow unit 12), does not extend to the ice shelf calving front.

**Mass balance**. Our results show that, on average, ice discharge from the Getz drainage basin is 116.4 Gt/year, with 23 Gt/year more ice flux in 2018 compared with 1994. The central region contributes the highest mass loss from the drainage basin partially due to it containing the fastest flowing and thickest ice. Cumulatively, the Getz study region has lost 315 Gt of ice mass since

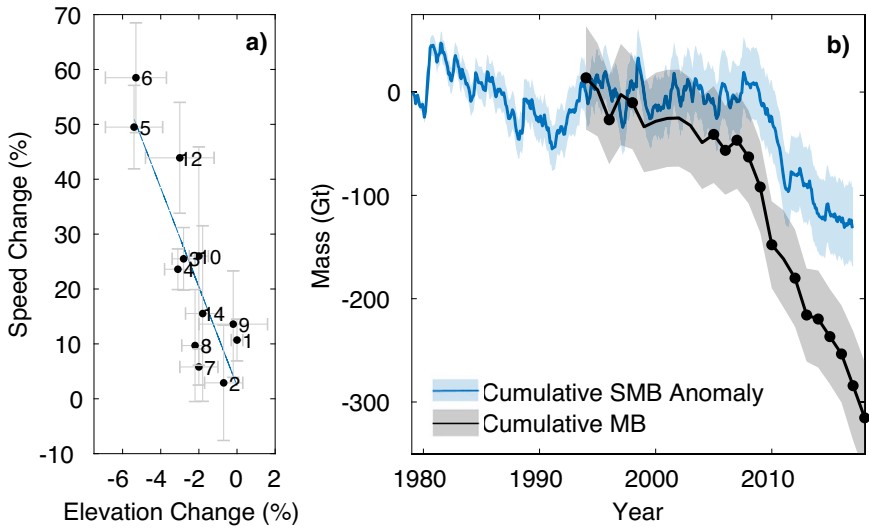

**Fig. 3 Ice dynamic relationship and mass balance. a** Total change in speed since 1994 as a percentage of 2018 velocity, scattered against change in ice thickness as a proportion of the total ice thickness. The number of each flow unit is annotated on each point, and the linear relationship (blue dashed line) between the two variables has an $R^2$ value of 0.74. Flow units 11 and 13 excluded due to low coverage of the velocity observations (<70%) in the 5 km region at the grounding line. The uncertainty bars (grey) are the standard deviation of the values extracted within the 5 km buffer region for each glacier. **b** The Cumulative Mass Balance from the Getz drainage basin between 1994 and 2018 (black line), filled black circles represent years with velocity observations, along with the cumulative Surface Mass Balance (SMB) anomaly from 1979 to 2016 (blue line)[68]. The mass balance uncertainty is calculated as the discharge error combined with the error in SMB. Discharge error is the combination of the percentage errors in ice thickness and velocity along the flux gate. For years of model discharge without a corresponding observation map, we assume the percentage error of the nearest observation year and add an additional standard deviation to the velocity error. We assume the uncertainty in the annual SMB trend to be 20% in line with previous studies[23,69] and the cumulative uncertainty is the root sum square of the annual errors, assuming that they are not correlated over time.

1994 with an uncertainty estimate of ±46 Gt/year, contributing 0.9 ± 0.6 mm global mean sea level equivalent over the last 25 years, with mass loss 4.4 times higher in the 2010s than it was in the 1990s (Fig. 3b). The SMB anomaly shows that following a short increase in surface mass in the early 1990s, the cumulative anomaly remained relatively constant from 1995 to 2008. From ~2008 to 2017, there was a substantial (~150 Gt) decrease in the cumulative SMB anomaly in the Getz basin, which is dominated by an anomalously low snowfall year in 2010 when snowfall was 57.3% lower than the annual mean over the 25-year period (103.8 Gt/year). Mass loss from the Getz drainage basin has therefore been caused by a linear increase in ice speed since the 1990s of 23.8%, exacerbated by a surface mass deficit in 2010. This shows the influence of both ice dynamic and surface mass processes in this region of Antarctica, with dynamic processes responsible for approximately two-thirds (57.4%) of the total mass loss. Our estimate of mass balance from the Getz sector using the input–output method compares well to other independent estimates of ice loss from the region[8,38,39], with differences attributed to mismatch between the precise study periods, spatial domain and ice thickness definitions.

## Discussion

Two previous studies have measured the change in speed of glaciers in the Getz drainage basin, with the largest increase in ice flow observed behind Siple Island and at the far West of the sector between 2007/8 to 2014/15 (refs. [23,36]). Our results extend and fill gaps in the spatial coverage, and show that the largest and most extensive ice speedup is concentrated on flow units 5 and 6 in the centre of the Getz drainage basin (Fig. 1b). Combined, these two glaciers account for 41.1% of the total increase in ice speed across the Getz sector since 1994 (Table 1). The zone of high percentage speedup coincides with the thickest and most rapidly thinning part of the Getz Ice Shelf behind Siple and Carney Islands[27,30]. This study confirms that the five fastest flowing glaciers in the far West of the Getz drainage basin (flow units 10 to 14) have also undergone a significant increase in ice speed since 1994 (~25% on average), accounting for 39.7% of the total speedup in the region (Table 1). DeVicq Glacier (flow unit 10) has been posed as a possible route through which future instability might propagate in Marie Byrd Land, due to a deep bedrock trough that lies over 300 m below sea level at the grounding line and extends over 200 km inland. Model studies have indicated that despite the glacier's geometry it is not susceptible to imbalance even with the presence of warmer ocean water in the Amundsen Sea[40]. In contrast, our results show that although limited to ~40 km inland of the grounding line, this glacier has undergone the fourth largest speed up in the Getz region over the last 25 years, with ice speeds increasing at a rate of 9.1 m/year$^2$ (Table 1).

Berry, Venzke and Land Glaciers (flow units 11, 12 and 14) correspond to a region of orographically driven high snowfall in excess of 3000 mm w.e./year, that is resolved by high spatial resolution (5.5 km) regional climate models[41]. The Getz drainage basin has the highest surface mass variability in Antarctica outside of the peninsula[8], and the cumulative surface mass anomaly shows that since ~2008 snowfall into the basin has been significantly lower than the long-term mean (Fig. 3b). Our multi-decadal time-series of speed change shows that on all glaciers in the Getz study region, the observed increase in ice flow has been relatively linear over the past 25 years (Supplementary Fig. 1d). The accelerated mass loss from the Getz sector since ~2010 observed by this study (Fig. 3b) and others[8,39], has therefore likely been driven by both the long-term gradual increase in dynamic imbalance combined with the effect of a short-term

surface mass deficit. This shows that extreme snowfall years have a significant influence on the mass balance of the whole drainage basin, and suggests that localised regions of high snowfall only resolved by high-resolution models may impact the balance of individual glaciers.

The Getz drainage basin is located in the transition zone between the cooler Ross Sea and the warmer Amundsen Sea, with warmer ocean potential temperatures found in the West of the Getz study region (Fig. 4c). Observations suggest that over the last three decades ocean temperatures in West Antarctica have warmed offshore[42], with periodic incursions of warm mCDW onto the continental shelf driving shorter term sub-decadal to decadal variability[14,15,25]. In the Getz region, annual mean temperature depth profiles collected from ship-based CTD sampling between 1994 and 2018, show annual variation in the depth of the thermocline dividing a cold and fresh upper layer from mCDW (Fig. 4b). In 1994, 2000, 2012 and 2014, the thermocline was ~200 m deeper, leading to cooler integrated ocean heat content, in contrast to warmer years in 2007, 2009, 2016 and 2018 when the thermocline was shallower. These observations are in line with previous studies of ocean temperature, and have been linked to the ocean response to interannual atmospheric variability[25,28]. Change in thermocline depth is observed as a band of temperature variability between ~500 and 800 m below sea level, and is particularly strong in the West of the Getz sector (Fig. 4d). The relatively short distance between the continental shelf break and the Getz calving front makes ocean heat content particularly sensitive to atmospheric forcing at the continental shelf break, even more so than in the Eastern Amundsen Sea[43,44], consistent with the large ocean heat content variability (Fig. 4b).

In order for spatial and temporal variability in ocean temperatures to affect the rate of ice melt and dynamic imbalance of the Getz drainage basin, warmer ocean water must be transported from the open ocean under the ice shelf at depth (below ~400 m), where it can come into contact with and melt deeply grounded ice. Ocean circulation under Antarctic ice shelves is directed by the Coriolis force, which causes water to enter ice cavities on the Eastern side before being guided by the seabed and ice base geometry and exiting on the Western side. On the Getz Ice Shelf, Coriolis-driven circulation (Fig. 4a) brings in warm ocean water from the Amundsen Sea between Wright and Duncan Islands, and to a lesser extent between Dean and Siple Islands. Warmer water travels along the grounding line towards the West, melting the ice and gaining buoyancy and upwelling with potential freshwater, along with buoyancy and circulation input from subglacial meltwater channels[45,46] (Fig. 4a). The pattern of ocean circulation under the Getz Ice Shelf is visible through higher meltwater fraction exiting the cavity at the ~150–200 m ice draft on the eastern side of islands, most clearly Wright, Duncan and Grant (Fig. 4e). We attribute the ice speedup observed on flow units 4 to 6 (Fig. 1b), in part to the impact of circulation-driven warm ocean water reaching the grounding line in these locations, driving high rates of ice speedup and ice shelf melt[27,30]. Relatively warmer water seems to reach the Western Getz[28] (Fig. 4). However, within the Getz cavities where tides appear to be relatively weak and melting relatively large, most of the ocean circulation is thought to be driven by the melt-induced upwelling and conducted by the evolving geometry, much like in Pine Island and Thwaites cavities[47]. In the case of Getz, its many ice stream tributaries, and islands acting as pinning points for the ice flow make for a complex and expectedly sinuous circulation pattern.

Ocean melt can also be enhanced by buoyancy gains at the glacier grounding zone provided by subglacial drainage outflows. Our results indicate that a number of glaciers in the Getz study region that have sped up coincide with regions of high sub-glacial water flux under the ice sheet[45] (Fig. 4a), including flow units 4 to

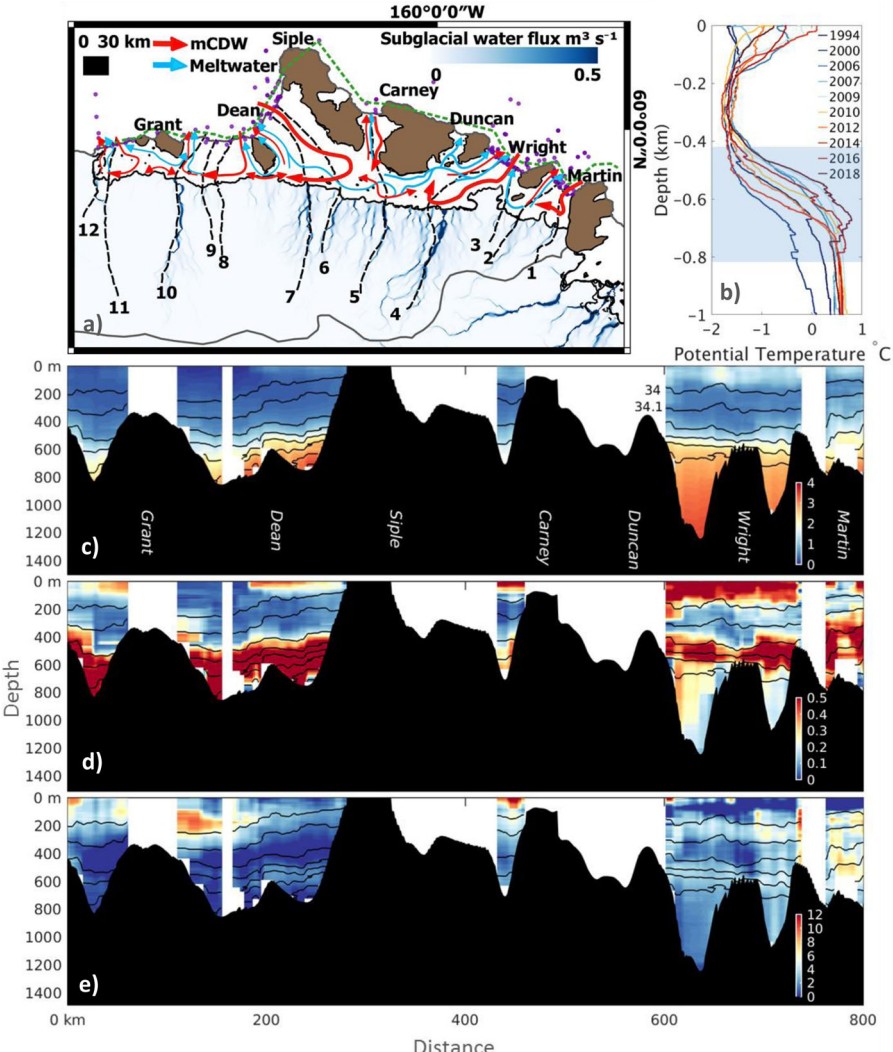

**Fig. 4 Ocean characteristics along the Getz Ice Shelf. a** Indicative ocean circulation under the Getz Ice Shelf cavity including both mCDW (red arrows) and meltwater (blue arrows); sub-glacial water flux under the ice sheet[45] (blue scale); the location of CTD sampling sites (purple dots) and the ocean transect (green dashed line) shown in **c**, **d** and **e**. The grounding line location (solid black line[67]), inland limit of the drainage basin (solid grey line) and the location of 14 flow line profiles (dashed black lines) are also shown. **b** Annual mean potential temperature as depth profiles from 1994 to 2018, from CTD measurements made at sampling sites (see **a**) along the Getz coastline. The shaded blue region indicates the depth of the grounding zone. **c** Average potential ocean temperature (°C) along a transect (see **a**) located at the calving front of the Getz Ice Shelf. Ocean salinity between 34 and 35 Practical Salinity Unit (PSU) (black contours at 0.1 PSU intervals) are also shown, along with the sea floor bathymetry (black shading[70]) with the names of large islands annotated. **d** Standard deviation of ocean potential temperature (°C) along the transect (see **a**), showing high (red) and low (blue) variability in ocean temperatures. **e** Meltwater Fraction (ml.l$^{-1}$) along the transect (see Fig. 4a) at the Getz Ice Shelf calving front.

6 and DeVicq Glacier (flow unit 10). Ice melt driven by sub-glacial runoff or, in the Amundsen sector ocean heat, is known to decrease with distance from the grounding line[14,45,48], as the entrainment of warm water at the ice–ocean interface increases with buoyancy and velocity, but diminishes downstream. In contrast, melt water channels under an ice shelf can extend to the ice front if the supply of warm ocean water is sustained[49], or if the features are simply advected downstream. Buoyant freshwater plumes are known to drive high melt rates on marine terminating ice[50,51], and studies have shown that on Getz Ice Shelf hydro-logical pathways routing subglacial freshwater play a significant role in determining the location of and rate of basal melt under the ice shelf[46]. The change in ice speed shows that in the Getz study region, increases are locally strongest near the grounding zone and do not extend to the ice front, indicating that subglacial hydrology could also be responsible for some of the observed speedup and thinning (Fig. 1b, c). If the volume of sub-glacial

water flux changes over time, as is the case on glaciers in Greenland due to seasonal surface melt[52,53], this may be an unaccounted-for factor driving change in ice flow and thinning in the regions of Antarctica with high sub-glacial water flux. Most ice flow models do not account for the presence of subglacial water and our knowledge about any change in subglacial water flux over the last 25 years is limited in regions not characterised by active sub-glacial lakes. The Getz drainage basin may therefore be a valuable test region for studies investigating the coupling between sub-glacial hydrology and the ocean, and the impact this may have on the localised pattern of ice sheet dynamics in Antarctica.

Our results show that on the majority of glaciers in the Getz drainage basin, ice speeds have increased at a broadly linear rate (Supplementary Fig. 1d). Given that the Getz basin, more so than its unstable neighbours in the Eastern Amundsen Sea, is thought to be relatively immune to positive feedback processes like MISI

and MICI due to its prograde bed topography, it is difficult to explain this trend as a runaway response to a step or oscillatory ocean forcing, at least during the past 25 years[15]. This suggests that the dynamic imbalance observed in Getz may primarily be a response to longer-term ocean forcing which may be anthropogenic in origin[16]. In the future, research programmes that deliver continuous annual monitoring of ice velocity and ocean temperatures across the study region at repeat locations will be critically important, preventing gaps in the record and enabling a more direct assessment of the link between the localised pattern and short-term variability of ice dynamics and the complex transport of ocean temperature variability.

Our 25-year-long record of ice speed shows for the first time that since 1994, widespread, linear speedup has occurred on the majority of glaciers in the Getz drainage basin of West Antarctica. Maps of change in ice flow show concentrated zones of very high speed up (over 44%) at the grounding line on three glaciers (flow units 6, 5, and Venzke Glacier (flow unit 12)), and high speed up of over 20% on an additional three glaciers (flow units 3, 4, DeVicq Glacier (flow unit 10)) since 1994. The central region of the Getz drainage basin accounts for 46.8% of this acceleration, and contains the most spatially extensive areas of change. The pattern of ice speedup indicates a localised response on individual glaciers, demonstrating the value of high-resolution observations that resolve the detailed pattern of dynamic imbalance across the Getz drainage basin. On all glaciers, the speed increase coincides with regions of high surface lowering, with an ~50% speed up corresponding to an ~5% reduction in ice thickness. Our optimised model results show that 315 Gt of ice has been lost from the Getz drainage basin since 1994, contributing $0.9 \pm 0.6$ mm to global sea levels, and increasing the rate of ice loss by four times in the 2010s compared to the 1990s. The prograde bed topography makes the Getz region inherently less susceptible to unstable geometry driven feedbacks such as MISI or MICI compared with its Amundsen Sea neighbours, and indicates that long-term ocean warming may be driving the dynamic imbalance in this region of Antarctica. Consistent and temporally extensive sampling of both ocean temperatures and ice speed will help further our understanding of dynamic imbalance in remote areas of Antarctica in the future.

## Methods

**Ice velocity observations**. We measured the ice speed of glaciers feeding the Getz Ice Shelf in West Antarctica over the last 25 years, using synthetic aperture radar (SAR) and optical satellite imagery acquired between January 1994 and December 2018 (Supplementary Table 1). Ice velocities were calculated using a combination of SAR and optical feature tracking[54,55] and SAR interferometry techniques[56,57]. To map ice velocity using the Sentinel-1a and -1b satellites, we tracked the displacement of features near to or on the ice surface such as rifts, crevasses and stable amplitude variations in SAR images[58,59]. We apply the intensity feature tracking technique to temporally sequential pairs of Interferometric Wide (IW) swath mode Single Look Complex (SLC) SAR images[60]. The image pairs are co-registered using a bilinear polynomial function constrained by precise orbital state vectors, resulting in a coregistration accuracy of 5 cm[60]. The normalised cross correlation of SAR intensity features is calculated over evenly spaced image patches, with a window size of 64 (~0.9 km) by 256 (~0.6 km) pixels in the azimuth and range directions, to compute the 2D offset for each patch[58,61]. The results are converted into absolute displacement in ground range coordinates using the Antarctic digital elevation model (DEM) posted on a 1 km grid[62]. The final velocity measurements are mosaicked and averaged to produce an annual ice speed map for each year of the study period.

Each velocity grid is post-processed to reduce noise and remove outliers by applying a low-pass (moving mean) filter over a 1 km by 1 km window, where values are rejected if the ice speed exceeds 30% of the mean value[60]. A second "dusting" step is applied to remove isolated pixels that are inconsistent with neighbouring velocity values, or where the measurements cover a region smaller than 0.1% (1.35 km$^2$) of the processed image size as sparse patchy data are an indicator of poor-quality measurements[60]. Errors in the velocity data are caused by imprecise co-registration of the SAR images, error in the auxiliary DEM, and atmospheric delay due to fluctuations in the ionosphere and tropospheric water vapour[35,61]. A spatially variable error measurement was computed for each velocity

grid by multiplying the signal-to-noise ratio of the cross-correlation function with the ice speed[60] (Supplementary Fig. 2). The largest velocity errors are found at the highly crevassed and rapidly deforming ice front and sheer margins of the glaciers, where under ~1.1% of the study region has a velocity error greater than 30%. Across the full Getz drainage basin, the mean velocity error for the 2017 and 2018 maps is 7.7% and 6.8%, respectively.

Historical Strip Map (SM) mode SLC SAR images acquired by the ERS-1 and ERS-2 satellites were used to measure ice speed in 1994, 1996 and 1998 using the intensity feature tracking technique[35,58] (Supplementary Table 1). MEaSUREs Antarctic annual ice velocity measurements were generated from a combination of SAR and optical Landsat-8 data using both feature tracking and interferometry techniques from data acquired over the period 2005 to 2017 (ref. [63]) (Supplementary Table 1). As with the Sentinel-1 velocity measurements, post processing filtering was applied to remove outliers where the error exceeded 0.3 m/d standard deviation and secondary "dusting" for the historical SAR data. The spatial resolution of the velocity maps ranges from 100 m to 1 km depending on the satellite imaging geometry, the window size used in the feature tracking step, and the DEM oversampling used in the geocoding step. Overall, this study presents the most temporally extensive record of ice velocity in the Getz drainage basin of West Antarctica, with 16 annual maps from 1994 to 2018 (Supplementary Fig. 2).

**Optimised ice flow model**. We used the BISICLES ice sheet model to interpolate velocity observations in time and space[35,64,65], calibrating the underlying model parameters—the vertically averaged viscosity and effective basal drag coefficients—and assuming that in the absence of observations they vary smoothly in both space and time (Supplementary Fig. 6). Scale distortions from the Polar Stereographic projection are not accounted for in the model output, therefore an additional systematic error of 1.7% is included within our discharge error estimate[23]. The observed and modelled ice speeds agree to within 11 m/year on average in all epochs (Supplementary Fig. 5a), and show the same broad spatial pattern of change in ice speed across the Getz basin (Supplementary Fig. 5b). Land Glacier and Hall Glacier (flow units 14 and 13) flow at the highest speeds, and flow units 5 and 6 have exhibited the largest percentage increase in ice speed since 1994 (Table 1). Interpretation of the underlying parameters must be treated with caution: the problem of their estimation is ill-posed and the solution depends on the choice of regularisation. Nonetheless the model is able to reproduce the observations without substantially altering the basal traction $\tau \lor$ (although the relevant model parameter, the effective basal drag coefficient $\tau \lor u$ does change substantially). In contrast, the vertically integrated effective viscosity does change, above and beyond the changes associated with the dependence of Glen's flow law on strain-rate, both in the ice shelf and in the fast-flowing streams. Supplementary Fig. 5d shows the relative change in a stiffening parameter ($\phi$) that quantifies this excess change. A reduction in $\phi$ can be interpreted as thinning, or damage (e.g. through crevassing). These results, then, though not the only possible interpretation, are consistent with acceleration caused by thinning of the ice shelf, resulting in crevasses opening upstream. At the same time, resistance from the bed does not increase (Supplementary Fig. 5c), consistent with a Coulomb friction law and with the behaviour of the trunk of Pine Island Glacier[66].

**Mass balance**. We calculated ice discharge from the Getz drainage basin using the 25-year record of optimised model ice flow (Supplementary Fig. 6) through a flux gate located at the grounding line[67], and a static ice sheet thickness from BEDMAP2 (ref. [32]). We investigated the impact of change in ice thickness on the ice discharge estimates, by reducing the ice thickness by 50 m at the flux gate. This is greater than the observed maximum thinning of 41.8 m over the full study period. We found that for 2018, this had a modest impact (1.5 %) on the total ice discharge. We assumed that the surface velocity in the direction normal to the flux gate equals the depth averaged speed with an assumed ice density of 917.2 kg/m$^3$. We computed mass balance over the 25-year study period by subtracting the ice flux out of the basin from the time-varying annual mean SMB, using a 27-km regional climate model (RACMO 2.3p2)[68]. The error for the total mass balance estimate arises from the errors in the discharge and SMB. Discharge error is the combination of the percentage errors in ice thickness and velocity along the flux gate. For years of model discharge without a corresponding observation map, we assume the percentage error of the nearest observation year and add an additional standard deviation to the velocity error. The mass balance uncertainty is calculated as the discharge error combined with the error in SMB. We assume the uncertainty in the annual SMB trend to be 20% in line with previous studies[23,69], and the cumulative uncertainty as the root sum square of the annual errors, assuming that they are not correlated over time. We calculated the SMB anomaly relative to the mean over the period 1979 to 2008, and accumulated over time[68] (Fig. 3b).

## Data availability

The Sentinel-1 2017 and 2018 velocity data that support the findings of this study are available from PANGAEA (https://doi.org/10.1594/PANGAEA.926520). The ERS-1 and -2 velocity data that support the findings of this study are available from the ENVEO Cryoportal, "Getz IV from ERS offset tracking, 1994", "Getz IV from ERS offset tracking, 1996" and "Getz IV from ERS offset-tracking, 1998" (http://cryoportal.enveo.at/data). The MEaSUREs annual velocity data that support the findings of this study are available from the NSIDC, "NSIDC-0720" (https://doi.org/10.5067/9T4EPQXTJYW9). The

optimised modelled velocity data that support the finding of this study are available from PANGAEA (https://doi.org/10.1594/PANGAEA.926520). The grounding line data that support the findings of this study are available from the European Space Agency (ESA) Climate Change Initiative (CCI) dashboard. The optimised ice sheet model that supports the findings of this study is available from BISICLES (https://commons.lbl.gov/display/bisicles/BISICLES). The CTD data that support the findings of this study are available from the National Oceanographic Data Center, "NBP9402", "NBP0001", "NBP0702", "NBP0901", "NBP1005" (https://www.ncei.noaa.gov/access/world-ocean-database-select/dbsearch.html), from PANGAEA, PS2006 (https://doi.org/10.1594/PANGAEA.847880) and PS2010 (https://doi.org/10.1594/PANGAEA.847944), from the British Oceanographic Data Centre, "JR141" (https://www.bodc.ac.uk/data/bodc_database/ctd/), from the Swedish National Data Service, "ODEN2007" (https://doi.org/10.5879/ECDS/2016-08-17.1/1), and from the Korea Polar Data Center, "ANA02C" (https://doi.org/10.22663/KOPRI-KPDC-00001176.4), "ANA04B" (https://doi.org/10.22663/KOPRI-KPDC-00000714.1), "ANA06B" (https://doi.org/10.22663/KOPRI-KPDC-00000636.1) and "ANA08B" (https://doi.org/10.22663/KOPRI-KPDC-00000907.1).

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

## Acknowledgements
This work was led by the School of Earth and Environment at the University of Leeds, with support from the NERC Centre for Polar Observation and Modelling (CPOM300001). The authors gratefully acknowledge the ESA, the National Aeronautics and Space Administration, the Japan Aerospace Exploration Agency and the Canadian Space Agency for the acquisition of ERS-1 and -2 (C1P9925), Sentinel-1, Landsat-8, ALOS PALSAR and RADARSAT data, respectively. We acknowledge the use of datasets produced through the ESA Antarctic Ice Sheet Climate Change Initiative (AIS_CCI) project and the NASA Measures programme for funding the development of long-term climate data records from satellite observations. Anna E. Hogg was supported by the NERC DeCAdeS project (NE/T012757/1) and ESA Polar+ Ice Shelves project (ESA-IPL-POE-EF-cb-LE-2019-834). Pierre Dutrieux was supported by NSF awards 1643285, 1644159, and a Columbia University Climate and Life Fellowship. Tae-Wan Kim from the Korea Polar Research Institute, grant KOPRI PE20160.

## Author contributions
H.L.S., A.E.H. and A.S. designed the work. H.L.S., A.E.H., J.W., T.N. and A.K. processed the ice velocity observations. S.C. performed the ice sheet model simulations. P.D. and T.W.K. provided oceanographic data. D.F. processed the grounding line observations. L.G. processed the surface elevation change observations. T.S. extracted surface mass balance data. H.L.S. performed the analysis. H.L.S. and A.E.H. wrote the manuscript. All authors contributed to scientific discussion, interpretation of the results and contributed to the manuscript.

## Competing interests
The authors declare no competing interests.
