## [Peer Review File · Nature Communications]

REVIEWER COMMENTS

Reviewer #1 (Remarks to the Author):

The authors use 25 years of surface velocity and surface mass balance data over the Getz region of West Antarctica. These data were used to infer a widespread increase in dynamical imbalance in this region. Although the dynamical imbalance in this region has been identified in many previous studies (e.g., Gardner et al., 2018, Rignot et al., 2019), the authors reinforce this with a long, high-quality, well-presented ice velocity record. The ice velocities were interpolated in space using an ice sheet model (similar to work by Minchew et al., 2018 – oddly not cited in this paper), which represents an important advance in the way these data are used and presented. My major concerns are only in the interpretation of the results, and not in the methodology.

Major comments:

1) Title: “Widespread increase in dynamic imbalance” is wrong if almost half of the mass loss since 2008 was driven by a decrease in surface mass balance? Perhaps the difference between total mass balance and surface mass balance anomaly (i.e., cumulative anomalies in dynamic mass balance) should be added to Figure 3 to illustrate this.

2) Figures S1 and S2 would be much more useful if they were anomalies relative to the 1994–2018 mean values. Right now, they’re pretty much the same figure repeated 16 times (I realize that potential sampling biases are highlighted in their present form, but they would be highlighted if the maps showed anomalies as well).

3) Line 136: I don’t see how a linear relationship confirms a lack of bias? You could have a linear relationship even if both the ice velocity data and surface elevation change data were biased.

4) Lines 201-217: I’m unsure about the relevance of this paragraph: besides the observed increase in ocean temperatures over the past three decades, the remaining changes in ocean conditions are at timescales much shorter than the ice velocity record is able to capture? I don’t see any correlation between temporal changes in ice sheet mass (Figure 3b) and ocean temperatures (Figure 4b), but they’re hard to interpret because of the different ways in which temporal information is presented in the two figures.

Mean ocean state isn’t as relevant to the authors’ claim of ocean-driven dynamic imbalance as a change in ocean heat content. Perhaps the authors can show the trend in ocean temperatures along a transect similar to Figures (c-e) and try to correlate those trends against the results in Figure 1b.

5) Lines 238-259: The authors dedicate considerable space to discuss the role of subglacial discharge on ice-sheet velocity change. However, I believe that the authors are interpreting correlations between subglacial discharge pathways and ice velocity changes as causation. The locations of subglacial discharge outflows are correlated with a variety of different variables, including ice velocity and ice thickness. Therefore, a correlation between an increase in ice velocity and average ice velocity or a correlation between an increase in ice velocity and ice thickness can manifest as a correlation between an increase in ice velocity and locations of subglacial discharge outflows.

This does not imply a causation: to show this, the authors must (ideally quantitatively) describe the physical mechanisms of how an unchanging rate of subglacial discharge can lead to their observed changes in ice velocity.

6) Ice thickness at the grounding line sometimes does not have the best observational constraints, which led to studies such as Gardner et al., 2018 to use other flux gate locations that had radar

data to constrain ice thickness. How does this affect the authors' results?

7) My preference would be to show the mass balance time series at the actual temporal sampling of the ice velocity data in Figure 3b, and not interpolated annual values.

Minor comments:

Line 31: mCDW melts floating ice, not grounded ice.

Line 54, Table 1: The uncertainties in this paper are very inconsistent: some uncertainties are provided, and sometimes not.

Line 55: Oceans in Antarctica -> Southern Ocean?

Line 91: Label all place names in Figures.

Lines 154-155: See comment (9) above.

Line 157: Gt/yr units are wrong for cumulative loss.

Line 161: "cumulative surface mass" -> "cumulative SMB anomaly"?

Line 176 vs. Line 181: Precise percentage in one but an approximate percentage (to the same number of significant digits!) in another?

Line 235: Rephrase. Ocean melt isn't "complemented" by buoyancy gains?

Line 291: Do the authors perform the flux gate calculations in PS71? If so, are scale distortions (which increase in magnitude from 71S) accounted for? Please explain in the Methods if they either are accounted for, or are not accounted for and represent a source of error in the processing.

Line 296: The SAR satellites are mentioned, but the optical satellites are not?

Figure S3: Should probably use $\phi_{2018} - \phi_{2007}$ instead of $\phi - \phi$ in the colorbar label. Similarly, for τ .

Figure S5,6: Mention whether elevations are relative to the geoid or the ellipsoid.

Figure 1 caption: I think the "D" in "FDM" is for "densification"?

Table 1: Add reference for grounding line data.

References:

Gardner, A. S. et al. Increased West Antarctic and unchanged East Antarctic ice discharge over the last 7 years. *Cryosphere* 12, 521–547 (2018).

Rignot, E., et al. "Four decades of Antarctic Ice Sheet mass balance from 1979–2017." *Proceedings of the National Academy of Sciences* 116.4 (2019): 1095-1103.

Minchew, B. M., et al. "Modeling the dynamic response of outlet glaciers to observed ice-shelf thinning in the Bellingshausen Sea Sector, West Antarctica." *Journal of Glaciology* 64.244 (2018): 333-342.

Reviewer #2 (Remarks to the Author):

Paper summary

The Getz region of Antarctica has become an increasingly large contributor to Antarctic ice mass losses over the last three decades. Despite this, there are relatively few in depth studies of the spatial and temporal evolution in ice dynamics and mass balance in the region, primarily due to difficulties faced by remote sensing observation systems. Therefore, in order to better understand the ice sheet contribution to sea level, there is a need for studies that look at the oceanographic and climate drivers in this region. This study therefore builds on previous literature over this sector of Antarctica and will be of interest to a cryosphere and sea level studies readership.

The study primarily focuses on an in-depth analysis of velocity observations, ranging from 1994 until 2018. These are used to provide evidence of a linear speed up in ice velocity across the major basin outlet glaciers, which when combined with ice sheet elevation change are used to make the case for dynamic thinning. The velocity observations are interpolated in time and space with the use of an ice sheet model, which allows for mass balance assessment to take place over the full 24-year period. These results support previous studies which have evidenced mass losses driven by ice dynamics and, particularly in the latter period, a reduction in SMB. The paper utilizes more recent comprehensive ice velocity products to achieve this, representing an advancement on previous studies which primarily looked at snapshots in time. It also complements other studies which have focused more on grounding line retreat (e.g. Christie et al 2018) and the ice shelf response to ocean/atmosphere forcing (e.g. Paolo et al 2018).

The paper is well written on the whole with extensive well-presented figures, well grounded in the literature and provides evidence for the major claims and paper outcomes. However, I think the paper text needs to be changed, particularly with the velocity observation analysis, to better reflect and represent the actual time period of data for which the majority of the data covers, or whether the analysis is using the modelled as well as observed velocities. For the majority of the flow unit outlets data only exists from 2005 onwards, whereas the discussion centers on speed up from 1990's. Whilst this claim is applicable for some of the central regions it cannot be applied everywhere based on observations alone. In addition, some improvements or clarification needs to be made to the mass balance calculation methodologies. With these improvements I believe it could be suitable for publication in a journal of Nature Communications scope. My major and specific comments on the text are provided below.

Major Comments

L107 - 109 How many of these sampling dates were actually in the 1990's though? From the looks of the coverage statistics in table S1 only a small number of the temporal samplings will be in the 1990's.

L113 – 119 - Whilst it is claimed in the text that the observed changes in ice speed cover the period from 1994 onwards, the majority of outlet flow units have observational coverage from 2005 onwards. Therefore, I don't believe you can say that the ice flow acceleration from observations alone is representative for the whole period with the use of a linear fit only covering part of the time series. The wording of this needs to be changed so that it is clearer which time period the trends calculated in figure S6d are representing or whether you are also using the modelled velocities in your analysis for this section.

The use of the ice sheet model to interpolate the velocities and the 1990's data is undoubtedly novel and useful for achieving the required coverage needed for mass balance calculations. The main issue is the presentation of the velocity observation data themselves covering the whole period, when in fact this is only the case for some of the locations at the Getz grounding line.

L114 – Relating to the point above, the only flow unit you can confidently in the 1990's in addition to rest of the time period based on observations alone is unit 10, the rest of them are based on observations going back to 2004/5 (figure S6d). Therefore, this needs to be clarified to better reflect the time period observed.

L118-L119 "we required all trends to use a minimum of 5-years of velocity observations" – Relating to the point above, is there also a minimum time span threshold for the 5 annual observations? If all observations are grouped in a single 5-year time span then the derived trend will not be indicative of the whole 1994-2018 period and may only capture a small snapshot of the dynamic change, which may be driven by enhanced by ocean warming over that period or sub-glacial melt water. I think this process needs to be better explained to the reader.

L120 – L121 Again see point above, this has actually accelerated from 2005 based on the data you have and what is presented in figure S6, it would not be appropriate to try and extrapolate that trend back to 1994 based on the observation data you have alone. Unless you are using the modelled ice velocities as well in this analysis.

L125 -126 Again the 0.7 m/yr² speed up you see on flow units 1 & 2 is based on a fit from 2005 until present, not 1994. Also going from the number of samples and R squared fit results for units 1 and 2 in figure S6d, I would be careful in the quantification of the 'acceleration'. The fitted trend would be susceptible to any inter-annual changes in ice velocity (e.g. the possibility of subglacial drainage enhanced flow you mention in the main text).

I think methodological improvements need to be made to the mass balance calculations in both the ice discharge and the SMB in order for them to be more in tune with the latest developments in the method:

L353 "flux gate located at the grounding line" this assumes that you have used a static grounding line that has not changed over the 25 year period? Christie et al (2018) shows non-trivial grounding line retreat of up to 1.5 km from 2003 until 2017, in addition to the grounding line retreat you claim in Table 1. Therefore, what impact would you expect this to have on the accuracy of your flux calculations? The grounding line files for Christie et al (2018) are available for the period 2003 until present (<https://doi.org/10.1594/PANGAEA.884780>) so this would be fairly easy to discover if it is a major or minor consideration. It could be that the resolution of Bedmap2 is the limiting factor for this, but I believe it should be considered.

L354 – A lot of the data used in Bedmap2 comes from older products, has there been an attempt to adjust ice thickness at the flux gate temporally in addition to the annual modelled ice velocities? You already used the elevation data from Shepherd et al (2019) in figure S6 so it should be a correction could be made at the grounding line. The impact could be small on the result, but it is something to consider and discuss, particularly when looking at analyses over multiple decades as you are in this case. As you show this change in figure S6b, if it has already been done already, clarification to the text should be made.

L361 – 363 – "The mass balance error is calculated as the discharge error combined with the error in SMB, assumed to be 5.9% in line with previous studies" – As far as I am aware the study you are referencing is not necessarily 'in line' with other previous studies in placing the SMB error at 5.9%. Other IOM studies have far larger SMB uncertainty assumptions, typically placing the SMB error at ~20% (Gardner et al., 2018; Wouters et al., 2015).

There is a lack of SMB observations to validate these models for much of West Antarctica, with none over the Getz study region (Melchior Van Wessem et al., 2018). In addition, when looking at SMB model observation comparisons, some of the largest uncertainties SMB model uncertainties are in low-lying coastal regions around the West Antarctic coast (Figure 5b of (Melchior Van Wessem et al., 2018)), for which the Getz is situated.

In lines 189-192 of the manuscript you raise the issue of the Getz needing high resolution climate models to resolve high snowfall regions and that it is a basin that has the highest spatial variability in snowfall outside the Antarctic Peninsula. Considering these statements, I believe a more conservative estimate of SMB uncertainty is required for this basin, more in the region of 20% as mentioned in previous studies, particularly as the 27 km model version is being used. This would bring your mass balance results more into the consensus with other IOM approaches. The 5.9% value does not seem suitable when looking at individual regional analyses, although I can understand why it was initially used and quantification of this uncertainty is challenging.

Specific Comments

L21 – “global mean sea levels” should be singular and not plural.

L25 – same as comment above.

L38-L39 - “surface mass observed since the 1990’s” This seems to imply that the surface mass balance estimates come from observations, when in fact they come from regional climate models. Unless the referenced article supporting this claim, which is not currently published, uses observations, this should be changed to better reflect the surface mass is from model outputs.

L53-54 – A recently released paper looking at basal melt rates across Antarctic ice shelves would also support your claim here. It was published after the submission of the manuscript but would be worthy of consideration:

Adusumilli, S., Fricker, H.A., Medley, B. et al. Interannual variations in meltwater input to the Southern Ocean from Antarctic ice shelves. *Nat. Geosci.* (2020). <https://doi.org/10.1038/s41561-020-0616-z>

L103 – 104 The percentage and areal coverage figures you are presenting here for 1998 do not match the values you are referring to in Table S1. Which is correct? Either the table or text needs to be changed with the correct value to ensure consistency.

L157 – “315.3 Gt/yr” – The decimal point precision is unnecessary as it very likely that the uncertainties in both ice thickness and velocity will mean that results at spatial scales such as the whole Getz sector cannot be precise to < 1 Gt

L303-304 – The DEM used to convert absolute displacement is quite old and has been superseded by products using better observations such as Bedmap2 (Fretwell et al., 2013), The Cryosat-2 DEM (Slater et al., 2018) and REMA (Howat et al., 2019). What is the impact of DEM accuracy on this process and its implication for introducing errors into the velocity observations?

Table 1 – The precision on the units for some variables seems to be higher than the actual stated precision of the products used. For example, ice thickness and bed elevation from Bedmap2 is provided at sub-meter precision, whereas the uncertainties in the product are much larger than this.

Table 1 – How are your grounding line migration values calculated? Is there a reference for this? Could the Christie et al 2018 data be used here to help supplement missing values? (even though it is only available from 2005 onwards)

Figure 1a – The white text on the greyscale topography background is quite difficult to read, it should be changed to ensure legibility.

Figure 4b – The blue shaded region in the plot refer to needs be explained in the caption for the figure.

Response to reviews of "Widespread increase in dynamic imbalance in the Getz region of Antarctica from 1994 to 2018"

Manuscript NCOMMS-20-24730

Response to Reviewers

We thank the reviewers and the editor for their time and effort in reviewing our paper, "Widespread increase in dynamic imbalance in the Getz region of Antarctica from 1994 to 2018", submitted for publication in Nature Communications. We welcome the positive feedback and insightful comments which we have endeavoured to fully address in this resubmitted revision, and we hope you agree this improves the manuscript. We have incorporated the majority of the suggestions made by the reviewers, and in the limited cases where we have not, we have provided a detailed description of the justification for each decision. The changes are highlighted in the manuscript. Please see below a point-by-point response to the reviewers' comments, where all line numbers refer to the revised manuscript file.

ID	Comment	Response
Reviewer #1		
1	Reviewer #1 (Remarks to the Author): The authors use 25 years of surface velocity and surface mass balance data over the Getz region of West Antarctica. These data were used to infer a widespread increase in dynamical imbalance in this region. Although the dynamical imbalance in this region has been identified in many previous studies (e.g., Gardner et al., 2018, Rignot et al., 2019), the authors reinforce this with a long, high-quality, well-presented ice velocity record. The ice velocities were interpolated in space using an ice sheet model (similar to work by Minchow et al., 2018 – oddly not cited in this paper), which represents an important advance in the way these data are used and presented. My major concerns are only in the interpretation of the results, and not in the methodology.	Done. The maximum number of references allowed in a Nature Communications article is 70, and given that we are at the limit of this allocation we aimed to prioritise the references we used most during the writing of this paper. We agree with the reviewers however that the Minchow et al., (2018) paper is highly relevant to this paper and was an omission. We removed previous reference 62, identifying the source of Sentinel-1 precise orbit information as this step in the methodology is already documented in the referenced Lemos et al., (2018) paper which is also cited in the methods text. We have cited Minchow et al., 2018, which is now referenced on Line 338.
2	Title: "Widespread increase in dynamic imbalance" is wrong if almost half of the mass loss since 2008 was driven by a decrease in surface mass balance? Perhaps the difference between total mass balance and surface mass balance anomaly (i.e., cumulative anomalies in dynamic mass balance) should be added to Figure 3 to illustrate this.	The term 'widespread' in the title is referring to the spatial pattern of dynamic imbalance across the Getz drainage basin. Figure 1b shows speedup and therefore increased dynamic ice loss across the length of the Getz coastline, impacting on all 14 glaciers in the region (Table 1). As this paper reports an increase in ice speed over the study period, the dynamic imbalance of the glaciers has also increased. This is true, even if in some more recent years (e.g. 2008) mass loss has also occurred due to anomalously low snowfall into the basin. It is for this reason that we continue to feel the wording of the title of the paper, 'Widespread increase in dynamic imbalance', is a justified and proportionate reflection of the results we present. The cumulative dynamic anomaly accounts for 57.4 % of the cumulative mass balance for the full study period, but per year the median dynamic anomaly is 70.6 %. This shows the large effect that one year's anomalous snowfall can have on multiple decades of total mass loss, but on an annual basis

		the dynamic imbalance is the dominant process at work in the Getz drainage basin. Edit Line 168-174: ‘Mass loss from the Getz drainage basin has therefore been caused by a linear increase in ice speed since the 1990’s of 23.8 %, exacerbated by a surface mass deficit in 2010. This shows the influence of both ice dynamic and surface mass processes in this region of Antarctica, with dynamic processes responsible for approximately two thirds (57.4 %) of the total mass loss.’
3	Figures S1 and S2 would be much more useful if they were anomalies relative to the 1994–2018 mean values. Right now, they’re pretty much the same figure repeated 16 times (I realize that potential sampling biases are highlighted in their present form, but they would be highlighted if the maps showed anomalies as well).	We agree with the reviewer that Figures S4 and S8 (previously S1 and S2) might appear repetitive as we present the annual velocity maps for each year of the study period. However, as this is the core dataset used in this paper and it is useful for readers to see both the spatial coverage and velocity magnitude for each year, we feel it is important to present our data in this form. If we were to present the velocity maps as an anomaly, it would reduce reader’s ability to directly compare our results to both past, and future, velocity studies of the region (e.g. Rignot et al, 2019, Gardner et al., 2018), therefore we have not changed the scaling to velocity anomaly relative to our mean time period.
4	Line 136: I don’t see how a linear relationship confirms a lack of bias? You could have a linear relationship even if both the ice velocity data and surface elevation change data were biased.	Done. It is possible that both datasets could be biased, resulting in a linear correlation. However, as the elevation change and ice velocity techniques are completely independent, measuring different parts related to the theory of mass conservation, we feel it is reasonable to point out that the correlation could indicate a lack of bias. We have amended the sentence to be less emphatic in its wording, see new sentence below: Edit Line 140-142: “This result indicates that a ~50 % increase in ice speed will cause a ~5 % decrease in glacier thickness due to ice dynamic processes, and suggests that there is no significant bias in the velocity or surface elevation change observations.”
5	Lines 201-217: I’m unsure about the relevance of this paragraph: besides the observed increase in ocean temperatures over the past three decades, the remaining changes in ocean conditions are at timescales much shorter than the ice velocity record is able to capture? I don’t see any correlation between temporal changes in ice sheet mass (Figure 3b) and ocean temperatures (Figure 4b), but they’re hard to interpret because of the different ways in which temporal information is presented in the two figures. Mean ocean state isn’t as relevant to the authors’ claim of ocean-driven dynamic imbalance as a change in ocean heat content. Perhaps the authors can show the trend in ocean temperatures along a transect similar to Figures (c-e) and try to correlate those trends against the results in Figure 1b.	The purpose of this paragraph is to present and describe the in-situ ocean data collected in and around the Getz drainage basin, during the study period. The paragraph provides important context about the Getz region and its surrounding ocean, and is the first time this ocean temperature data has been published in their entirety, (Jacobs et al., (2013) who we reference covers some of the period). As ocean forcing is thought to be the primary driver of dynamic imbalance in Antarctica, it is important to present the change in ocean temperature around Getz, as if present, it is a possible cause of the change in ice speed results presented earlier in the paper. Before examining the data, we did not know if there was any variability in the ocean temperature around Getz, and as the reviewer agrees, this paragraph reports that there is short-term variability in ocean heat (Lines 223-242). It is of interest that the timing of the observed ocean temperature variability is consistent with short-term temperature variability in the neighbouring Amundsen Sea (Dutrieux et al 2014; Jenkins et al., 2018), but as the reviewer points out we have not reported a direct correlation between the short term changes in ocean temperature and short term ice speed in the Getz basin. On lines 271 - 275 we say that more and

		consistently sampled ocean data are needed and hopefully this study will help highlight the importance of this work. Given the amplitude of the variability, the ocean dataset is unfortunately too short to identify eventual trends. Therefore, we have stopped short of explicitly stating that there isn't a direct correlation, because the focus of this paper was on generating a long term, multi-decade ice speed record, and as we have edited later on in the paper (Lines 271-275) to say we think that a dedicated study investigating short term variability would be of great value. Edit Line 271-275: 'In the future, research programmes that deliver continuous annual monitoring of ice velocity and ocean temperatures across the study region at repeat locations will be critically important, preventing gaps in the record and enabling a more direct assessment of the link between the localised pattern and short-term variability of ice dynamics and the complex transport of ocean temperature variability.'
6	Lines 238-259: The authors dedicate considerable space to discuss the role of subglacial discharge on ice-sheet velocity change. However, I believe that the authors are interpreting correlations between subglacial discharge pathways and ice velocity changes as causation. The locations of subglacial discharge outflows are correlated with a variety of different variables, including ice velocity and ice thickness. Therefore, a correlation between an increase in ice velocity and average ice velocity or a correlation between an increase in ice velocity and ice thickness can manifest as a correlation between an increase in ice velocity and locations of subglacial discharge outflows. This does not imply a causation: to show this, the authors must (ideally quantitatively) describe the physical mechanisms of how an unchanging rate of subglacial discharge can lead to their observed changes in ice velocity.	As the reviewer notes, in the discussions section of this paper we consider the possibility that sub-glacial hydrological pathways may have a role to play in influencing the change in ice speed observed in the Getz region. Throughout the majority of this paragraph (Lines 243 to 251) we set out information reported in previous publications, (e.g. Le Brocq et al., 2013) freshwater input to the ocean drives plumes (Jenkins et al., 2011; Slater et al., 2016) which cause higher melt rates primarily at the grounding line; ii) sub-glacial hydrological pathways cause enhanced basal melt on the Getz ice shelf (Le Brocq et al., 2013; Wei et al., 2020), and we qualitatively discuss how an understanding of our results could be informed by these processes. As the reviewer states we have not quantitatively demonstrated a causation, primarily because a change in the sub-glacial hydrology would be required to drive a change in ice speed (not just the presence of a hydrological pathway), and as we also note in the manuscript (Lines 259 to 261), the variability of subglacial discharge under Antarctica is completely unknown, apart from in the case of active subglacial lakes. Within the context of paper discussions, it is valid to suggest with appropriate caveats, (lines 256-259) that "If the volume of sub-glacial water flux changes over time, as is the case on glaciers in Greenland due to seasonal surface melt (Sundal et al., 2011; Nienow et al., 2017), this may be an unaccounted-for factor driving change in ice flow and thinning in the regions of Antarctica with high sub-glacial water flux,"; and to suggest that Getz would be a valuable test region for a dedicated study on this topic in the future (Lines 261 to 264).
7	Ice thickness at the grounding line sometimes does not have the best observational constraints, which led to studies such as Gardner et al., 2018 to use other flux gate locations that had radar data to constrain ice thickness. How does this affect the authors' results?	The best and most recent estimate of ice thickness near the grounding line in the Getz basin is an Operation Ice Bridge Flight line that was acquired in 2016 (see map below).

Unfortunately, this transect doesn't cover the full margin of the Getz basin and is on occasion located on the ice shelf, therefore our assessment is that the grounding line is a better location for the flux gate. We compared the ice thickness of Bedmap2 to the OIB ice thickness measurements, (see image below), which show the same broad spatial pattern of surface and bed height indicating the comparable nature of both datasets. The difference between both datasets is dominated by the bed topography component, and we primarily attribute this to the spatial resolution of both datasets, with Bedmap2 gridded up at coarser spatial resolution. Previous studies that have tested using multiple (5) flux gate locations, showed a small change in the mass balance estimate (3 Gt/yr range) (Mouginot et al. 2014), suggesting that changing the location of the flux gate would have only modest impact.

We also tested moving the grounding line inland by reducing grounding line thickness by 50 m, retreating the grounding line by approximately 5 km. Accounting for the 50 m of ice thickness changes causes a 2 Gt/yr increase in ice discharge in 2018. This is a modest impact which is 1.5 % of the total ice discharge for 2018. The maximum observed rate of thinning along the Getz grounding line is -1.7 m/a, which over 25 years gives us a total ice thinning of 41.8 m. This is less than with our assumed 50 m/a thinning rate sensitivity test, therefore the modelled impact of 2 Gt/yr is conservative, and well within our uncertainty estimate. Ultimately our tests show that changing the flux gate location would not significantly change the ice discharge findings reported in this study.

8 My preference would be to show the mass balance time series at the actual temporal sampling of the ice velocity data in Figure 3b, and not interpolated annual values.

Done. We have clarified the years where ice velocity observations exist in Figure 3b, by using a circle marker for years with observations. A time evolving ice flow model was used to calculate mass balance in years without observations, as this has the benefit of making use of all available data while presenting our most accurate estimate in years where observations were not acquired. See new plot below:

		 Figure 3 consists of two panels. Panel (a) is a scatter plot with 'Speed Change (%)' on the y-axis (0 to 60) and 'Elevation Change (%)' on the x-axis (-6 to 0). Data points are numbered 1 through 14. Panel (b) is a line graph with 'Mass (Gt)' on the y-axis (-300 to 0) and 'Year' on the x-axis (1980 to 2016). It shows a blue line for 'Cumulative SMB Anomaly' and a black line with filled circles for 'Cumulative MB'. Shaded areas represent uncertainty bounds.
9	Line 31: mCDW melts floating ice, not grounded ice.	Done. Edit Line 31: '(mCDW) melting the floating ice'
10	Line 54, Table 1: The uncertainties in this paper are very inconsistent: some uncertainties are provided, and sometimes not.	We provide a spatially and temporally varying error estimate for all our velocity products, which are the core new dataset presented in this paper. We also provide an uncertainty estimate for our mass balance estimate. However, some of the previously published datasets, such as the elevation change with firn densification correction, don't have an associated uncertainty product available. We are not able to provide uncertainties for these auxiliary products.
11	Line 55: Oceans in Antarctica -> Southern Ocean?	Done. Edit Line 55: 'largest sources of fresh water input to the Southern Ocean'
12	Line 91: Label all place names in Figures.	Done. All place names mentioned in text are now added to Figure 1. We have not added the individual glacier names onto the five named glaciers to avoid hiding the data below and for consistency as we number the flow lines in the following figures for ease of recognition throughout the text.
13	Lines 154-155: See comment (10) above.	Done. See comment 10 response.
14	Line 157: Gt/yr units are wrong for cumulative loss.	Done. Edit Line 161: '315 Gt of ice mass since 1994 with an uncertainty estimate of ± 41 Gt/yr'. Edit Line 287-288: 'Our optimised model results show that 315 Gt of ice has been lost from the Getz drainage basin since 1994'
15	Line 161: "cumulative surface mass" -> "cumulative SMB anomaly"?	Done Edit Line 166: 'cumulative SMB anomaly'.
16	Line 176 vs. Line 181: Precise percentage in one but an approximate percentage (to the same number of significant digits!) in another?	Done. We have changed the level of significant digits for the approximate percentage. Edit on Line 185: '(~25 % on average)'
17	Line 235: Rephrase. Ocean melt isn't "complemented" by buoyancy gains?	Done. We have removed 'complemented' from the sentence. Edit on line 243: "Ocean melt can also be enhanced by buoyancy gains at the glacier grounding zone provided by subglacial drainage outflows."
18	Line 291: Do the authors perform the flux gate calculations in PS71? If so, are scale distortions (which increase in magnitude	Done. We did not account for the projection distortion in the original flux estimate which is output by the model on a polar-stereo 71° grid. The grounding line at Getz is ~75

	from 71S) accounted for? Please explain in the Methods if they either are accounted for, or are not accounted for and represent a source of error in the processing.	degrees south, where the scale factor is 1.017 (Snyder, 1987). We have now accounted for the projection distortion within our error estimate using a systematic error of 1.7% of discharge as also done by previous studies, i.e. Gardner et al., (2018). We have updated our methods text to reflect this. We have also updated Figure 3 to reflect this (please see below).  Figure 3 consists of two panels. Panel (a) is a scatter plot with 'Speed Change (%)' on the y-axis (0 to 60) and 'Elevation Change (%)' on the x-axis (-6 to 0). Data points are numbered 1 through 14. A dashed line represents a linear fit to the data. Panel (b) is a line graph with 'Mass (Gt)' on the y-axis (-350 to 50) and 'Year' on the x-axis (1980 to 2015). It shows two data series: 'Cumulative SMB Anomaly' (blue line with a light blue shaded uncertainty region) and 'Cumulative MB' (black line with a grey shaded uncertainty region). Both series show a general downward trend over time, with the cumulative mass balance reaching approximately -350 Gt by 2015. Edit on Line 341-343: Scale distortions from the Polar Stereographic projection are not accounted for in the model output, therefore an additional systematic error of 1.7 % is included within our discharge error estimate (Gardner et al., 2018). Edit on Line 161: ± 46 Gt/yr
19	Line 296: The SAR satellites are mentioned, but the optical satellites are not?	We focus on a detailed description of the Sentinel SAR satellite methodology as this was newly developed for this study. The methods used to process the ERS SAR data and Landsat-8 optical data has been reported in previous publications (Mouginot et al., 2014; Strozzi et al., 2002) so we haven't repeated this. Table S1 also outlines the various data sources including the optical satellites.
20	Figure S3: Should probably use $\phi_{2018} - \phi_{2007}$ instead of $\phi - \phi$ in the colorbar label. Similarly, for τ.	Done. The figure has been amended as suggested, see below.

21 Figure S5,6: Mention whether elevations are relative to the geoid or the ellipsoid.

Done. The BEDMAP2 and Shepherd et al. (2019) products are referenced to the Polar stereographic projection, WGS84 ellipsoid, with true scale 71°S. This has been added to the figure captions for S6, S2 and S4 (previously S4, S5 and S6).

Edit Figure S6 caption: ‘The ice surface elevation (grey line) and bed elevation (brown line), in Polar stereographic projection referenced to the WGS84 ellipsoid, extracted along the same profile from BEDMAP2 (Fretwell et al. 2013) is also shown’.

Edit Figure S2 caption: ‘The ice surface elevation (grey line) and bed elevation (brown line), in Polar stereographic projection referenced to the WGS84 ellipsoid, extracted along the same profile from BEDMAP2 (Fretwell et al. 2013) is also shown’.

Edit Figure S4 caption: ‘The surface elevation change measured is relative to the WGS84 ellipsoid from radar altimetry data’

22 Figure 1 caption: I think the “D” in “FDM” is for “densification”?

Done.
Edit Figure 1 caption: ‘densification’.

23 Table 1: Add reference for grounding line data.

The grounding line data used created by Dana Floricioiu who is a co-author on the manuscript. The data was generated as

		part of the European Space Agency (ESA) Climate Change Initiative (CCI), with all datasets freely available through the CCI data portal or on request. We have added it to our data availability statement (see comment 46).
Reviewer #2		
24	Reviewer #2 (Remarks to the Author): The paper is well written on the whole with extensive well-presented figures, well grounded in the literature and provides evidence for the major claims and paper outcomes. However, I think the paper text needs to be changed, particularly with the velocity observation analysis, to better reflect and represent the actual time period of data for which the majority of the data covers, or whether the analysis is using the modelled as well as observed velocities. For the majority of the flow unit outlets data only exists from 2005 onwards, whereas the discussion centers on speed up from 1990's. Whilst this claim is applicable for some of the central regions it cannot be applied everywhere based on observations alone. In addition, some improvements or clarification needs to be made to the mass balance calculation methodologies. With these improvements I believe it could be suitable for publication in a journal of Nature Communications scope. My major and specific comments on the text are provided below.	We thank the reviewer for their positive comments, and have responded to the specific requests for changes in the appropriate spot below.
25	L107 - 109 How many of these sampling dates were actually in the 1990's though? From the looks of the coverage statistics in table S1 only a small number of the temporal samplings will be in the 1990's.	Done. On Lines 108 to 112 we acknowledge the limited spatial coverage of the velocity observations from the 1990s, however this data provides an invaluable reference measurement for glaciers in the central to Western part of the Getz sector, where no other ice speed estimate is available. We also show the temporal sampling of the speeds at the grounding line for each glacier in Figure S6 panel d and the spatial distribution of all velocity measurements in Figure S1.
26	L113 – 119 - Whilst it is claimed in the text that the observed changes in ice speed cover the period from 1994 onwards, the majority of outlet flow units have observational coverage from 2005 onwards. Therefore, I don't believe you can say that the ice flow acceleration from observations alone is representative for the whole period with the use of a linear fit only covering part of the time series. The wording of this needs to be changed so that it is clearer which time period the trends calculated in figure S6d are representing or whether you are also	Done. See answer to point 25 above. We acknowledge the paucity of data across the study region from the 1990's, however by using the optimised model approach we have developed a physically meaningful mechanism of interpolating and extrapolating data to regions with no observations. The plot below shows the linear trend for the change in ice speed on each glacier in the region. We have plotted data from our observations (black) and the model solution (blue) which shows good agreement between the two datasets on all glaciers, including those with no observations in the 1990's. We specifically used a linear fit to the data rather than a higher order polynomial regression, to ensure our estimates in the absence of observations remained conservative. We think that linear regression

using the modelled velocities in your analysis for this section.

The use of the ice sheet model to interpolate the velocities and the 1990's data is undoubtedly novel and useful for achieving the required coverage needed for mass balance calculations. The main issue is the presentation of the velocity observation data themselves covering the whole period, when in fact this is only the case for some of the locations at the Getz grounding line.

remains the best fit possible suggest they are representative of the full 25-year study period. We have added this new figure to the supplementary information (Figure S5) to illustrate the similarity in trends.

We have also added a sentence to clarify that the trends do not all necessarily include data from the 1990s. The edit has been copied below also:

Edit Line 114-118: 'Our results show that all glaciers in the study area have accelerated since the 1990's, with 7 glaciers speeding up by over 20 % (flow units 3 to 6, 10, 12 and 13) (Table 1) based on a linear rate (Figures S6d, S8 and S4). In regions without observational data from the 1990's this trend may not be tightly constrained by the data, however, we find good agreement between the modelled and observed rates (Figure S5).'

Edit Figure S5 caption: 'Change in ice speed for each flow unit across the Getz Ice Basin. The observations (black circles) and the optimized model velocities (blue triangles) are shown throughout the study period. The linear fit to both datasets is also show, with good agreement between the two.'

27 L114 – Relating to the point above, the only flow unit you can confidently in the 1990's in addition to rest of the time period based on observations alone is unit 10, the rest of them are based on observations going back to 2004/5 (figure S6d). Therefore, this needs to be clarified to better reflect the time period observed.

Please see response to points 25 and 26 above.

28 L118-L119 "we required all trends to use a minimum of 5-years of velocity observations" – Relating to the point above, is there also a minimum time span threshold for the 5 annual observations? If all observations are grouped in a single 5-year time span then the derived trend will not be indicative of the whole 1994-2018 period and may only capture a small snapshot of the dynamic change, which may be driven by enhanced by ocean warming over that period or sub-glacial melt water. I think this process needs to be better explained to the reader.

We currently do not apply a minimum time span threshold but merely require a minimum of 5 data points to ensure a valid trend could be calculated in each pixel. As we state on line 122 we also apply a 95 % confidence interval filter. It is true that each pixel fit represents different time periods covered but we have an average of 10 years' worth of data points for each flow line and we are confident in the trends observed as outlined above in response 25. I have clarified this in text and the edit has been copied below also:

Line 121-122: 'We required all trends to include a minimum of 5-years of velocity observations, not necessarily consecutively, and we filtered the output using a 95% confidence interval on the fit.'

29	L120 – L121 Again see point above, this has actually accelerated from 2005 based on the data you have and what is presented in figure S6, it would not be appropriate to try and extrapolate that trend back to 1994 based on the observation data you have alone. Unless you are using the modelled ice velocities as well in this analysis.	Please see response to points 25 and 26 above. We used a linear fit to the data rather than a higher order polynomial regression, to ensure our estimates in the absence of observations remained conservative. A detailed study of the short-term velocity variability would be a really interesting future study.
30	L125 -126 Again the 0.7 m/yr ² speed up you see on flow units 1 & 2 is based on a fit from 2005 until present, not 1994. Also going from the number of samples and R squared fit results for units 1 and 2 in figure S6d, I would be careful in the quantification of the ‘acceleration’. The fitted trend would be susceptible to any inter-annual changes in ice velocity (e.g. the possibility of subglacial drainage enhanced flow you mention in the main text).	Please see response to points 25 and 26 above. Our calculated rate for flow units 1 and 2 agrees particularly well with the modelled trend for those glaciers (Figure S5).
31	I think methodological improvements need to be made to the mass balance calculations in both the ice discharge and the SMB in order for them to be more in tune with the latest developments in the method:	Please see below our response to specific comments on MB and SMB methods.
32	L353 “flux gate located at the grounding line” this assumes that you have used a static grounding line that has not changed over the 25 year period? Christie et al (2018) shows non-trivial grounding line retreat of up to 1.5 km from 2003 until 2017, in addition to the grounding line retreat you claim in Table 1. Therefore, what impact would you expect this to have on the accuracy of your flux calculations? The grounding line files for Christie et al (2018) are available for the period 2003 until present (https://doi.org/10.1594/PANGAEA.884780) so this would be fairly easy to discover if it is a major or minor consideration. It could be that the resolution of Bedmap2 is the limiting factor for this, but I believe it should be considered.	As the reviewer notes, we use a static grounding line position (Rignot et al., 2016) as the flux gate location in this study. In order to calculate ice flux across the Getz basin, we required a complete gate that crosses the full margin of the basin. Neither the Christie et al. (2018) Grounding Line dataset, or the ESA CCI dataset we present in this paper are complete. Christie et al has a large gap across DeVicq Glacier (flow unit 13), and the ESA CCI datasets are also only partial. Therefore the Rignot et al. (2016) dataset remains the best estimate of the grounding line when a complete boundary is required. The resolution of the BISICLES ice flow model output is 1 km, which means that even in locations where grounding line migration has occurred this would result in a maximum of 1 pixel difference if a different location was used. It is worth noting that the Christie et al. (2018) dataset does not show uniform grounding line retreat across the basin, a significant proportion of the margin has experienced no retreat (See map below).

33	L354 – A lot of the data used in Bedmap2 comes from older products, has there been an attempt to adjust ice thickness at the flux gate temporally in addition to the annual modelled ice velocities? You already used the elevation data from Shepherd et al (2019) in figure S6 so it should be a correction could be made at the grounding line. The impact could be small on the result, but it is something to consider and discuss, particularly when looking at analyses over multiple decades as you are in this case. As you show this change in figure S6b, if it has already been done already, clarification to the text should be made.	We agree with the reviewer that ice thickness can affect the flux estimates. The original flux estimate did not account for thickness changes. We have investigated its impact by reducing the grounding line thickness by 50 m, which retreats the grounding line by approximately 5 km in our model simulations. Accounting for 50 m of ice thickness change causes a 2 Gt/yr increase in ice discharge in 2018. This is a modest impact which is 1.5 % of the total ice discharge for 2018. The maximum observed rate of thinning along the Getz grounding line is -1.7 m/yr, which over 25 years gives us a total ice thinning of 41.8 m. This is less than with our assumed 50 m/yr thinning rate sensitivity test, therefore the modelled impact of 2 Gt/yr is conservative, and well within our uncertainty estimate. Ultimately our tests show that correcting for thickness change would not significantly change the ice discharge findings reported in this study.
34	L361 – 363 – “The mass balance error is calculated as the discharge error combined with the error in SMB, assumed to be 5.9% in line with previous studies” – As far as I am aware the study you are referencing is not necessarily ‘in line’ with other previous studies in placing the SMB error at 5.9%. Other IOM studies have far larger SMB uncertainty assumptions, typically placing the SMB error at ~20% (Gardner et al., 2018; Wouters et al., 2015). There is a lack of SMB observations to validate these models for much of West Antarctica, with none over the Getz study region (Melchior Van Wessem et al., 2018). In addition, when looking at SMB model observation comparisons, some of the largest uncertainties SMB model uncertainties are in low-lying coastal regions around the West Antarctic coast (Figure 5b of (Melchior Van Wessem et al., 2018)), for which the Getz is situated. In lines 189-192 of the manuscript you raise the issue of the Getz needing high resolution climate models to resolve high snowfall regions and that it is a basin that has the highest spatial variability in snowfall outside the Antarctic Peninsula. Considering these statements, I believe a more conservative estimate of SMB uncertainty is required for this basin, more in the region of 20% as mentioned in previous studies, particularly as the 27 km model version is being used. This would bring your mass balance results more into the consensus with other IOM approaches. The 5.9% value does not	Done. We have adjusted our SMB uncertainty to be 20 % which widens our uncertainty range in Figure 3b (see below). The uncertainty estimates for the mass balance have been updated throughout the manuscript, the sea level equivalent uncertainty remains the same at ± 0.6 mm. This includes the additional uncertainty from reviewer comment 18.  Figure 3b consists of two panels, (a) and (b). Panel (a) is a scatter plot with 'Speed Change (%)' on the y-axis (ranging from 0 to 60) and 'Elevation Change (%)' on the x-axis (ranging from -6 to 0). There are 12 data points, each labeled with a number from 1 to 12. A dashed line represents a linear regression fit to the data. Panel (b) is a line graph with 'Mass (Gt)' on the y-axis (ranging from -350 to 50) and 'Year' on the x-axis (ranging from 1980 to 2015). It shows two data series: 'Cumulative SMB Anomaly' (represented by a blue line with a light blue shaded uncertainty range) and 'Cumulative MB' (represented by a black line with a grey shaded uncertainty range). Both series show a general downward trend over time, with the Cumulative MB showing a steeper decline after 2000. Edit line 370-374: ‘The mass balance uncertainty is calculated as the discharge error combined with the error in SMB. We assume the uncertainty in the annual SMB trend to be 20 % in line with previous studies (Gardner et al., 2018; Wouters et al., 2015), and the cumulative uncertainty as the root sum square of the annual errors, assuming that they are not correlated over time.’ Edit Line 161: ‘with an uncertainty estimate of ± 46 Gt/yr’

	seem suitable when looking at individual regional analyses, although I can understand why it was initially used and quantification of this uncertainty is challenging.	
35	L21 – “global mean sea levels” should be singular and not plural.	Done. Edit Line 21: ‘global mean sea level’
36	L25 – same as comment above.	Done. Edit Line 25: ‘global mean sea level’
37	L38-L39 - “surface mass observed since the 1990’s” This seems to imply that the surface mass balance estimates come from observations, when in fact they come from regional climate models. Unless the referenced article supporting this claim, which is not currently published, uses observations, this should be changed to better reflect the surface mass is from model outputs.	Done. Word ‘observed’ deleted from the sentence. Edit Line 37-39 : ‘Satellite data has shown that in Antarctica the dynamic ice loss (6.3 ± 1.9 mm sea level equivalent (sle)) is 86 % greater than the modest reduction in surface mass (0.9 ± 1.1 mm sle) since the 1990’s (Slater et al., 2020).’
38	L53-54 – A recently released paper looking at basal melt rates across Antarctic ice shelves would also support your claim here. It was published after the submission of the manuscript but would be worthy of consideration: Adusumilli, S., Fricker, H.A., Medley, B. et al. Interannual variations in meltwater input to the Southern Ocean from Antarctic ice shelves. Nat. Geosci. (2020). https://doi.org/10.1038/s41561-020-0616-z	We agree with the reviewer that the newly published Adusumilli et al. (2020) paper is highly relevant to our study, however, as we are at the limit of the number of references allowed by Nature communications, we would have to remove an existing reference in order to make space. As Susheel’s paper was only just published while our existing manuscript was under review, the results of this paper ultimately didn’t inform our study. We look forward to discussing the complimentary nature of both papers with Susheel and his co-authors at the next conference we all attend, and I have no doubt that his paper will be widely cited. We feel it wouldn’t be right to remove a reference that was used to inform our methods and discussion, to make room to cite a paper that wasn’t available to us at the time of writing.
39	L103 – 104 The percentage and areal coverage figures you are presenting here for 1998 do not match the values you are referring to in Table S1. Which is correct? Either the table or text needs to be changed with the correct value to ensure consistency.	Done. The values have been corrected in text to match the table which are the filtered velocity output coverage rather than the raw values that the previous text referred too. Edit Line 105: ‘2 % ($3,432 \text{ km}^2$)’
40	L157 – “315.3 Gt/yr” – The decimal point precision is unnecessary as it very likely that the uncertainties in both ice thickness and velocity will mean that results at spatial scales such as the whole Getz sector cannot be precise to < 1 Gt	Done. We agree, and have removed the decimal point precision. Edit Line 161: ‘315 Gt’ Edit Line 20: ‘315 Gt of ice has been lost contributing 0.9 ± 0.6 mm’ Edit Line 287: ‘315 Gt’
41	L303-304 – The DEM used to convert absolute displacement is quite old and has been superseded by products using better observations such as Bedmap2 (Fretwell et al., 2013), The Cryosat-2 DEM (Slater et al., 2018) and REMA (Howat et al., 2019). What is the impact of DEM accuracy on this process and its implication for	Although the DEM used can have a limited impact on the accuracy of the velocity products, the associated error is lowest in regions of low slope and will vary over large spatial scales. It was out of the scope of this study to do a full sensitivity test to investigate the impact of different DEM products, however we agree with the reviewer that this could be an interesting topic for a future technical ice velocity paper to investigate in detail.

	introducing errors into the velocity observations?																																																																																																																																																							
42	Table 1 – The precision on the units for some variables seems to be higher than the actual stated precision of the products used. For example, ice thickness and bed elevation from Bedmap2 is provided at sub-meter precision, whereas the uncertainties in the product are much larger than this	Done. The precision of units have been changed to better represent the product precision. The amended Table 1 is copied below:    Flow Unit 2018 Mean Speed (m/yr) Observed Rate of Speed Change (m/yr²) Modelled rate of Speed Change (m/yr²) Observed Total Speed up over 25 years (m/yr) Observed Percentage Speed Change over 25-years (%) Rate of Elevation Change 1992 - 2017 (m/yr) Grounding Line Migration 1996-2017 (km) Ice thickness (m) Bed Elevation (m)    1173.7 ± 11.50.70.818.610.70.0-1.3649-540 2192.2 ± 17.20.20.15.52.9-0.2-1.1674-594 3182.6 ± 14.41.91.946.525.5-0.7-2.7639-541 4390.7 ± 33.33.73.492.423.6-1.0-775-651 5532.4 ± 36.110.59.1263.449.5-1.6-728-613 6668.8 ± 90.815.615.2391.258.5-1.8-851-727 7424.8 ± 42.21.02.124.55.8-0.5-2.5638-552 8518.8 ± 59.12.05.450.49.7-0.5-501-418 9511.2 ± 64.62.83.869.413.6-0.1-552-473 10879.2 ± 149.19.111.4228.226.0-0.6-659-535 11744.3 ± 125.05.75.1141.519.0-2.2-913-659 12467.8 ± 65.58.26.2205.543.9-1.3-995-806 131397.6 ± 163.711.46.0285.820.4-0.4-674-524 141281.1 ± 122.68.07.3198.915.5-0.6-701-560  	Flow Unit	2018 Mean Speed (m/yr)	Observed Rate of Speed Change (m/yr ²)	Modelled rate of Speed Change (m/yr ²)	Observed Total Speed up over 25 years (m/yr)	Observed Percentage Speed Change over 25-years (%)	Rate of Elevation Change 1992 - 2017 (m/yr)	Grounding Line Migration 1996-2017 (km)	Ice thickness (m)	Bed Elevation (m)	1	173.7 ± 11.5	0.7	0.8	18.6	10.7	0.0	-1.3	649	-540	2	192.2 ± 17.2	0.2	0.1	5.5	2.9	-0.2	-1.1	674	-594	3	182.6 ± 14.4	1.9	1.9	46.5	25.5	-0.7	-2.7	639	-541	4	390.7 ± 33.3	3.7	3.4	92.4	23.6	-1.0	-	775	-651	5	532.4 ± 36.1	10.5	9.1	263.4	49.5	-1.6	-	728	-613	6	668.8 ± 90.8	15.6	15.2	391.2	58.5	-1.8	-	851	-727	7	424.8 ± 42.2	1.0	2.1	24.5	5.8	-0.5	-2.5	638	-552	8	518.8 ± 59.1	2.0	5.4	50.4	9.7	-0.5	-	501	-418	9	511.2 ± 64.6	2.8	3.8	69.4	13.6	-0.1	-	552	-473	10	879.2 ± 149.1	9.1	11.4	228.2	26.0	-0.6	-	659	-535	11	744.3 ± 125.0	5.7	5.1	141.5	19.0	-2.2	-	913	-659	12	467.8 ± 65.5	8.2	6.2	205.5	43.9	-1.3	-	995	-806	13	1397.6 ± 163.7	11.4	6.0	285.8	20.4	-0.4	-	674	-524	14	1281.1 ± 122.6	8.0	7.3	198.9	15.5	-0.6	-	701	-560
Flow Unit	2018 Mean Speed (m/yr)	Observed Rate of Speed Change (m/yr ²)	Modelled rate of Speed Change (m/yr ²)	Observed Total Speed up over 25 years (m/yr)	Observed Percentage Speed Change over 25-years (%)	Rate of Elevation Change 1992 - 2017 (m/yr)	Grounding Line Migration 1996-2017 (km)	Ice thickness (m)	Bed Elevation (m)																																																																																																																																															
1	173.7 ± 11.5	0.7	0.8	18.6	10.7	0.0	-1.3	649	-540																																																																																																																																															
2	192.2 ± 17.2	0.2	0.1	5.5	2.9	-0.2	-1.1	674	-594																																																																																																																																															
3	182.6 ± 14.4	1.9	1.9	46.5	25.5	-0.7	-2.7	639	-541																																																																																																																																															
4	390.7 ± 33.3	3.7	3.4	92.4	23.6	-1.0	-	775	-651																																																																																																																																															
5	532.4 ± 36.1	10.5	9.1	263.4	49.5	-1.6	-	728	-613																																																																																																																																															
6	668.8 ± 90.8	15.6	15.2	391.2	58.5	-1.8	-	851	-727																																																																																																																																															
7	424.8 ± 42.2	1.0	2.1	24.5	5.8	-0.5	-2.5	638	-552																																																																																																																																															
8	518.8 ± 59.1	2.0	5.4	50.4	9.7	-0.5	-	501	-418																																																																																																																																															
9	511.2 ± 64.6	2.8	3.8	69.4	13.6	-0.1	-	552	-473																																																																																																																																															
10	879.2 ± 149.1	9.1	11.4	228.2	26.0	-0.6	-	659	-535																																																																																																																																															
11	744.3 ± 125.0	5.7	5.1	141.5	19.0	-2.2	-	913	-659																																																																																																																																															
12	467.8 ± 65.5	8.2	6.2	205.5	43.9	-1.3	-	995	-806																																																																																																																																															
13	1397.6 ± 163.7	11.4	6.0	285.8	20.4	-0.4	-	674	-524																																																																																																																																															
14	1281.1 ± 122.6	8.0	7.3	198.9	15.5	-0.6	-	701	-560																																																																																																																																															
43	Table 1 – How are your grounding line migration values calculated? Is there a reference for this? Could the Christie et al 2018 data be used here to help supplement missing values? (even though it is only available from 2005 onwards)	Done. The grounding line migration number in Table 1 was calculated as the distance the grounding line moved along the central flow line go each glacier. This has been clarified in the table caption. The data used is the European Space Agency (ESA) Climate Change Initiative (CCI) grounding line product generated by Dana Floricioiu, a co-author on the manuscript. The Christie et al. (2018) dataset is extremely valuable, however, the use of optical shadow rather than InSAR methods would require us to do a full inter-comparison between the datasets to ensure any change was due to real movement of this feature, rather than differences in methodology. It was beyond the scope of this paper to do that task. Edit Table 1 caption: “Grounding line migration is calculated as the distance along the central flow line the grounding line has moved, negative indicating a movement inland.”																																																																																																																																																						
44	Figure 1a – The white text on the greyscale topography background is quite difficult to read, it should be changed to ensure legibility.	Done. We changed the white text on Figure 1a and 1b to black for better visibility. The edited figure is below:																																																																																																																																																						

45 Figure 4b – The blue shaded region in the plot refer to needs be explained in the caption for the figure.

Done. The shaded blue region description has been added to the figure 4b caption.
 Edit Figure 4 caption: 'b) Annual mean potential temperature as depth profiles from 1994 to 2018, from CTD measurements made at sampling sites (see a) along the Getz coastline. The shaded blue region indicates the depth of the grounding zone.'

46 In particular, both reviewers mentioned to us (and we agree with them) that they think your data should be made publicly available to the community alongside your manuscript. Please see below about our data policies and how to submit your data either to a repository or in a Supplementary Data (e.g. Excel or .txt) file.

We have changed our data availability statement as suggested and have chosen to make the core velocity datasets available. We have submitted our Sentinel-1 dataset to Pangea and it is in the process of being placed in the repository, however this means we are yet to receive a DOI. Similarly the CCI grounding line data will be available shortly through the ESA CCI dashboard. We have also linked the other two velocity datasets.

Edit Line 377-382: The Sentinel-1 2017 and 2018 velocity data that support the findings of this study are available from PANGAEA. The ERS-1 & -2 velocity data that support the findings of this study are available from the ENVEO Cryportal, <http://cryportal.enveo.at/data>. The MEaSURES annual velocity data that support the findings of this study are available from the NSIDC, <http://nsidc.org/data/measures>. The grounding line data that

	support the findings of this study are available from the European Space Agency (ESA) Climate Change Initiative (CCI) dashboard.
--	--

References

- Adusumilli, S., Fricker, H. A., Medley, B., Padman, L. & Siegfried, M. R. Interannual variations in meltwater input to the Southern Ocean from Antarctic ice shelves. *Nat. Geosci.* 1–5 (2020). doi:10.1038/s41561-020-0616-z
- Christie, F. D. W. *et al.* Glacier change along West Antarctica’s Marie Byrd Land Sector and links to inter-decadal atmosphere–ocean variability. *Cryosph.* **12**, 2461–2479 (2018).
- Dutrieux, P. *et al.* Strong sensitivity of pine Island ice-shelf melting to climatic variability. *Science (80-.).* **343**, 174–178 (2014).
- Gardner, A. S. *et al.* Increased West Antarctic and unchanged East Antarctic ice discharge over the last 7 years. *Cryosphere* **12**, 521–547 (2018).
- Jacobs, J. *et al.* Cryptic sub-ice geology revealed by a U-Pb zircon study of glacial till in Dronning Maud Land, East Antarctica. *Precambrian Res.* **294**, 1–14 (2017).
- Jenkins, A. *et al.* West Antarctic Ice Sheet retreat in the Amundsen Sea driven by decadal oceanic variability. *Nat. Geosci.* **11**, 733–738 (2018).
- Le Brocq, A. M. *et al.* Evidence from ice shelves for channelized meltwater flow beneath the Antarctic Ice Sheet. *Nat. Geosci.* (2013). doi:10.1038/NCEO1977
- Nienow, P. W., Sole, A. J., Slater, D. A. & Cowton, T. R. Recent Advances in Our Understanding of the Role of Meltwater in the Greenland Ice Sheet System. *Current Climate Change Reports* **3**, 330–344 (2017).
- Minchew, B. M., Gudmundsson, G. H., Gardner, A. S., Paolo, F. S. & Fricker, H. A. Modeling the dynamic response of outlet glaciers to observed ice-shelf thinning in the Bellingshausen Sea Sector, West Antarctica. *J. Glaciol.* **64**, 333–342 (2018).
- Mouginot, J., Rignot, E. & Scheuchl, B. Sustained increase in ice discharge from the Amundsen Sea Embayment, West Antarctica, from 1973 to 2013. *Geophys. Res. Lett.* **41**, 1576–1584 (2014).
- Rignot, E. *et al.* Four decades of Antarctic ice sheet mass balance from 1979–2017. *Proceedings of the National Academy of Sciences of the United States of America* **116**, 1095–1103 (2019).
- Rignot, E., Mouginot, J., B. S. MEaSURES Antarctic Grounding Line from Differential Satellite Radar Interferometry, Version 2. Boulder, Color. USA. NASA Natl. Snow Ice Data Cent. Distrib. Act. Arch. Center. Available from <https://doi.org/10.5067/IKBWW4RYHF1Q.02/04/2018> (2016).
- Sundal, A. V. *et al.* Melt-induced speed-up of Greenland ice sheet offset by efficient subglacial drainage. *Nature* **469**, 521–524 (2011).
- Slater, D. A., Goldberg, D. N., Nienow, P. W. & Cowton, T. R. Scalings for Submarine Melting at Tidewater Glaciers from Buoyant Plume Theory. *J. Phys. Oceanogr.* **46**, 1839–1855 (2016).
- Slater, T., Hogg, A. E. & Mottram, R. Ice-sheet losses track high-end sea-level rise projections. *Nature Climate Change* **10**, 879–881 (2020).
- Snyder, J. P. *Map Projection: A Working Manual (U.S. Geological Survey Professional Paper 1395). Professional Paper* (U.S. Government Printing Office, 1987). doi:10.3133/pp1395
- Strozzi, T., Luckman, A., Murray, T., Wegmüller, U. & Werner, C. L. Glacier motion estimation using SAR offset-tracking procedures. *IEEE Trans. Geosci. Remote Sens.* **40**, 2384–2391 (2002).
- Wei, W. *et al.* Getz Ice Shelf melt enhanced by freshwater discharge from beneath the West Antarctic Ice Sheet. *Cryosph.* **14**, 1399–1408 (2020).
- Wouters, B. *et al.* Dynamic thinning of glaciers on the Southern Antarctic Peninsula. *Science (80-.).* **348**, 899–903 (2015).

REVIEWERS' COMMENTS

Reviewer #2 (Remarks to the Author):

I thank the authors for addressing the comments in the original revision round and providing detailed responses to the questions asked. The majority of the major concerns have been addressed, particularly regarding the SMB model uncertainties and their propagation into the uncertainties. With regards to the observational time series not being fully representative of 1994 – 2018 time period, my concerns have been allayed by the general agreement between the modelled and observed velocity regression.

There are still a couple of issues in my opinion which I think need addressing but otherwise would be suitable for publication. See comments below:

Comment ID 30 – I still don't agree with the comment regarding the significance of the acceleration and don't believe this has been addressed. Whilst the modelled and observed velocities may agree, when you look at the trends in Fig S3D, the trend fits on the observations have R2 values of 0.01 and 0.02, with a fitted trend on a 12 data points. For example, the large negative 2009 value in Flow unit 1 can have a large influence on any fitted trend given the relatively few data points available. Therefore, I don't think you can confidently call it an acceleration within the uncertainties of the observations or modelled values you have (or there is large natural inter-annual variability). I understand the fact that the authors only have a limited amount of data, however this conclusion seems inappropriate based on the statistical analysis undertaken.

Comment ID 33 - I appreciate the authors explanation of the limited impact thickness change has on the discharge estimates, I think this should be added to the main text or supplement methods just to explain this to the reader. The full explanation in the rebuttal isn't necessarily, just a statement of its limited percentage impact.

Comment ID 46 - With regards to the data availability, I think the modelled velocities should also be made available as it is a core component of the analysis.

Response to reviews and editorial comments of "Widespread increase in dynamic imbalance in the Getz region of Antarctica from 1994 to 2018"

Manuscript NCOMMS-20-24730

Response to Reviewers

We thank the reviewers and the editor for their time and effort in reviewing our paper, "Widespread increase in dynamic imbalance in the Getz region of Antarctica from 1994 to 2018", submitted for publication in Nature Communications. We welcome the further feedback on our manuscript, we have incorporated the majority of the suggestions made by the reviewers, and the changes are highlighted in the manuscript. Please see below for a point-by-point response to the reviewers' comments, where all line numbers refer to the revised manuscript file.

ID	Comment	Response
Reviewer 2		
30	I still don't agree with the comment regarding the significance of the acceleration and don't believe this has been addressed. Whilst the modelled and observed velocities may agree, when you look at the trends in Fig S3D, the trend fits on the observations have R2 values of 0.01 and 0.02, with a fitted trend on a 12 data points. For example, the large negative 2009 value in Flow unit 1 can have a large influence on any fitted trend given the relatively few data points available. Therefore, I don't think you can confidently call it an acceleration within the uncertainties of the observations or modelled values you have (or there is large natural inter-annual variability). I understand the fact that the authors only have a limited amount of data, however this conclusion seems inappropriate based on the statistical analysis undertaken.	Done. We still feel that our results show clear speedup across the Getz basin, but we have toned down the language used to reflect the lower statistical correlation of flow unit 1 and 2 as suggested by the reviewer. Edit Line 118: "Our results show that the majority of glaciers in the study area have accelerated since the 1990's, with 7 glaciers speeding up by over 20 % (flow units 3 to 6, 10, 12 and 13) (Table 1) based on a linear rate (Supplementary Figs. 1d, 3 and 4)." Edit Line 269: "Our results show that on the majority of glaciers in the Getz drainage basin ice speeds have increased at a broadly linear rate (Supplementary Fig. 3d)." Edit Line 281: "Our 25-year long record of ice speed shows for the first time that since 1994, widespread, linear speedup has occurred on the majority of glaciers in the Getz drainage basin of West Antarctica."
33	I appreciate the authors explanation of the limited impact thickness change has on the discharge estimates, I think this should be added to the main text or supplement methods just to explain this to the reader. The full explanation in the rebuttal isn't necessarily, just a statement of its limited percentage impact.	Done. We have added a sentence into the methods to state the sensitivity of our results to thickness changes. Edit Line 366: We investigated the impact of change in ice thickness on the ice discharge estimates, by reducing the ice thickness by 50 m at the flux gate. This is greater than the observed maximum thinning of 41.8 m over the full study period. We found that for 2018,

		this had a modest impact (1.5 %) on the total ice discharge.
46	With regards to the data availability, I think the modelled velocities should also be made available as it is a core component of the analysis.	Done. We have changed our data availability statement as suggested and have chosen to make the modelled velocity datasets available. We have submitted our modelled dataset to Pangea and it is in the process of being placed in the repository, however this means we are yet to receive a DOI. Edit Line 388: The optimized modelled velocity data that supports the findings of this study are available from PANGAEA.